# Latent Trajectory Learning for Limited Timestamps under Distribution Shift over Time

**Qiuhao Zeng**[1]  **Changjian Shui**[2]  **Long-Kai Huang**  **Peng Liu**[3]  **Xi Chen**[4]
**Charles X. Ling**[1]  **Boyu Wang**[1]*
[1]University of Western Ontario  [2]Vector Institute  [3]University of Toronto  [4]Noah's Ark Lab

## Abstract

Distribution shifts over time are common in real-world machine-learning applications. This scenario is formulated as Evolving Domain Generalization (EDG), where models aim to generalize well to unseen target domains in a time-varying system by learning and leveraging the underlying evolving pattern of the distribution shifts across domains. However, existing methods encounter challenges due to the limited number of timestamps (every domain corresponds to a timestamp) in EDG datasets, leading to difficulties in capturing evolving dynamics and risking overfitting to the sparse timestamps, which hampers their generalization and adaptability to new tasks. To address this limitation, we propose a novel approach *SDE-EDG* that collects the **I**nfinitely **F**ined-**G**rid **E**volving **T**rajectory (IFGET) of the data distribution with continuous-interpolated samples to bridge temporal gaps (intervals between two successive timestamps). Furthermore, by leveraging the inherent capacity of Stochastic Differential Equations (SDEs) to capture continuous trajectories, we propose their use to align SDE-modeled trajectories with IFGET across domains, thus enabling the capture of evolving distribution trends. We evaluate our approach on several benchmark datasets and demonstrate that it can achieve superior performance compared to existing state-of-the-art methods.

## 1 Introduction

Domain Generalization (DG) is a fundamental problem in machine learning. It aims to learn a model that can perform well on unseen data based on the knowledge learned from multiple related domains (Arjovsky et al., 2019; Li et al., 2018b; Sagawa et al.; 2019; Sun & Saenko, 2016; Chen et al., 2024). However, traditional DG techniques assume that the distribution in different domains remains stationary over time, which is often impractical in many real-world scenarios. In practice, the distribution of data may shift over time due to factors such as changes in the environment or the underlying system. For example, Age-related changes occur in all ocular tissues, including age-related structural changes in the optic nerve (Grossniklaus et al., 2013). Age-related ocular disease is the most prevalent condition associated with vision impairment and blindness in older adults worldwide (Flaxman et al., 2017). However, the data collected from individuals aged beyond 80 is lacking due to a very small sample size, privacy, and other factors. It's necessary to build a prediction model based on the age-related pattern learned from the data collected from younger cases.

To adapt to changing environments over time, recent research in the community has focused on the scenario of Evolving Domain Generalization (EDG) (Bai et al., 2023; Nasery et al., 2021b; Qin et al., 2022; Zeng et al., 2023; Yao et al., 2022), aimed to tackle such problems. Specifically, the goal of EDG is to learn and leverage the evolving patterns captured from source domains to achieve generalization capability on the unseen future target domains in a time-varying environment.

However, one fundamental obstacle in existing EDG studies (Bai et al., 2023; Qin et al., 2022; Zeng et al., 2023) is that they suffer from a limited number of timestamps, resulting in overfitting to the sparse timestamp data. Consequently, they cannot properly capture the underlying evolving pattern and extrapolate into the future. In fact, recent research (Mariet & Kuznetsov, 2019) has revealed that

---

*Corresponding authors: Boyu Wang, Charles X. Ling.

the sample complexity of time series forecasting tasks scales as $\mathcal{O}(\sqrt{1/M})$, where $M$ is the number of timestamps of the training time-series data.

In this paper, we tackle this problem by constructing a continuously evolving trajectory. Specifically, we create the Infinitely Fined-Grid Evolving Trajectory (IFGET) in the latent representation space, with two steps: i) firstly, we develop sample-to-sample correspondence to collect the evolving trajectory of each individual sample; ii) Next, we generate continuous-interpolated samples by leveraging such correspondence, aimed to bridge the temporal gaps between timestamp intervals and avoid overfitting to sparse timestamps. It is denoted as IFGET, since it is a continuous trajectory and thereby can be subdivided into infinitely fine temporal grids. Nevertheless, dealing with continuous trajectories poses another challenge in EDG. Most existing EDG algorithms are designed for discrete timestamps and are not able to handle continuous timestamps, since they employ transition functions to predict data at the next timestamp based on current observation, inherently representing time discretely (Bai et al., 2023; Qin et al., 2022; Zeng et al., 2023; 2024).

To address this issue, we propose to model the temporal dynamics of latent representations by employing Stochastic Differential Equations (SDEs) (Kong et al., 2020; Li et al., 2020; Xu et al., 2022; Kidger et al., 2021) to fit the IFGET, which provides a natural approach for characterizing continuous-time stochastic processes. Specifically, we propose a *path alignment regularizer*, which aligns the latent trajectories characterized by SDEs with the paths generated by IFGET, by maximizing the likelihood of the SDE trajectories based on the observations of IFGET.

To summarize, our proposed algorithm, termed as SDE-EDG, has the following desirable properties[1]:

**Capturing Evolving Patterns via Infinitely Fined-grid Evolving Trajectory (IFGET)** To overcome the limitations of the small number of timestamps in current EDG data, we propose to learn the evolving dynamics by constructing the IFGET. To construct the evolving trajectory, we identify the sample-to-sample correspondence between successive domains and employ an interpolation function to generate a continuous-interpolated sample. IFGET alleviates overfitting to the limited timestamps and improves the generalization to distribution shifts over time.

**Modelling Trajectories of Latent Representations with Stochastic Differential Equations (SDEs)** Leveraging the inherent capacity of SDE to model continuous temporal trajectory, our model aligns the depicted latent trajectories of SDE-EDG with IFGET during the training process. We show that SDE-EDG is capable of quantifying evolving stochastic processes, and theoretically demonstrate that SDE-EDG results in a lower generalization bound for downstream tasks.

## 2 RELATED WORKS

**Evolving Domain Adaptation / Generalization** Evolving Domain Adaptation (EDA) (Hoffman et al., 2014; Kumagai & Iwata, 2017; Mancini et al., 2019; Liu et al., 2020a; Wang et al., 2020; Kumar et al., 2020; Wang et al., 2022) is a related field that focuses on scenarios where a single labeled domain is available alongside multiple unlabeled intermediate domains. The objective of EDA is to achieve generalization on unseen target domains. Recently, EDG has received considerable attention from researchers. Approaches to solving the EDG problem can be broadly categorized into two groups. The first group parameterizes the learning model with time-sensitive models (Qin et al., 2022; Nasery et al., 2021a; Bai et al., 2023). (Qin et al., 2022) proposes to tackle the challenges of covariate shift and concept shift, with the probabilistic model incorporating variational inference. The second group (Zeng et al., 2023; 2024) maps the source data into future data leveraging evolving patterns. However, these methods still suffer from limited timestamps, which hinder the capture of temporal trends. In contrast, we propose to construct IFGET, where the temporal gaps are filled with interpolations.

**Ordinary Differential Equations (ODE)/ Stochastic Differential Equations (SDE)** In recent years, Neural-ODEs (Chen et al., 2018b; Sun et al., 2020) have emerged as a powerful tool for continuous-time representation of neural networks. SDEs (Øksendal & Øksendal, 2003; Kong et al., 2020; Liu et al., 2020b) have incorporated stochastic terms into ODE solvers, injecting the model with slight random noise to improve generalization ability and noise robustness. To learn SDE neural networks' parameters, (Ryder et al., 2018; Xu et al., 2022; Li et al., 2020) use variational inference

---

[1]Our code is available at `https://github.com/HardworkingPearl/SDE-EDG-official`.

technology and overcome the overfitting problem. SDE has shown excellent performance in machine learning applications, such as generative adversarial model (GAN) (Kidger et al., 2021; Park et al., 2021a), score-based diffusion model (Song et al., 2020). In this work, we propose a novel approach to modeling Evolving Domain Generalization (EDG) as a dynamical system by utilizing SDEs to effectively represent the continuously evolving trajectory of the data representations, and we apply maximum likelihood to efficiently fit the latent evolving trajectories utilizing SDEs.

## 3 Preliminaries

**Ordinary / Stochastic Differential Equations**   Neural ordinary differential equations (Neural ODEs) (Chen et al., 2018a) approximate the evolving dynamics with the ordinary differential equation and are defined as

$$z_t = z_0 + \int_0^t f(z_s, s)\mathrm{d}s, \tag{1}$$

where the hidden state $z_t \in \mathbb{R}^d$ evolves with certain dynamics characterized by a neural network $f : \mathbb{R}^d \to \mathbb{R}^d$, $z_0$ is the initial state, and $s$ represents time in integrals. An SDE can be regarded as an ODE injected with noise over time:

$$z_t = z_0 + \int_0^t f(z_s, s)\mathrm{d}s + \int_0^t g(z_s, s)\mathrm{d}B_s, \tag{2}$$

where $z_t$ is a latent state that evolves over time, $f : \mathbb{R}^d \times \mathbb{R} \to \mathbb{R}^d$ is the drift function to capture the evolving dynamics, $g : \mathbb{R}^d \times \mathbb{R} \to \mathbb{R}^{d \times \omega}$ is the diffusion funtion to reflect the uncertainties, and $B_s$ is an $d$-dimensional Brownian motion (Wiener Process). SDE has shown superior performance in modeling the dynamical system (Park et al., 2021a;b; Øksendal & Øksendal, 2003). Under the EDG settings, the drift function of SDE describes the trends of the distribution shift over time, and the diffusion function models the samples' individual stochastics in their representation space.

**Evolving Domain Generalization**   Let $\mathcal{D}(x, y, t)$ be the probability distribution that characterizes temporal dynamics of an instance $x \in \mathcal{X}$ and its label $y \in \mathcal{Y}$, in which there exist underlying evolving patterns over $t$. In Evolving Domain Generalization (EDG), we are given $M$ source domains $\{S_m\}_{m=1}^M$, where $S_m = \{(x_{i|m}, y_{i|m})\}_{i=1}^N$ is the data of $m$-th domain sampled from $\mathcal{D}(x, y|t_m)$ at the timestamp $t_m \in [0, T]$, and $N$ is the sample size of the $m$-th domain. Note that most existing works (Bai et al., 2023; Nasery et al., 2021b; Qin et al., 2022; Zeng et al., 2023) assume a constant interval $\Delta t$ between two consecutive domains. In contrast, our approach exhibits flexibility in tackling the EDG problem even with irregular time intervals. In the proposed method, we will learn the evolving dynamics in a latent space $\mathcal{Z}$, given the practical advantages (Kirchmeyer et al., 2022), e.g., improved discrimination, dimension reduction, and computation resources savings compared to operations in the original input space $\mathcal{X}$. Specifically, for every instance $x$, we encode it with a feature extractor $\phi : \mathcal{X} \to \mathcal{Z}$, and we obtain the embedded feature $z = \phi(x) \in \mathcal{Z}$. We focus on the dynamics of $\mathcal{D}(z, y, t)$ throughout this work.

The goal of EDG is to learn a robust and generalizable model from source domains by capturing and leveraging the evolving pattern learned from source domains so that it can perform well on the unseen target domains in the path space (Boué & Dupuis, 1998) at $L$ timestamps $t_{M+l} \in \{t_{M+1}, \ldots, t_{M+L}\} \in (T, T + T^*]$ $(T^* = t_{M+L} - T)$:

$$\min_\theta R_\nu(h_\theta) = \min_\theta \mathbb{E}_{(z,y) \sim \mathcal{D}(z_{M+1:M+L}, y_{M+1:M+L})}[h_\theta(z) \neq y], \tag{3}$$

where $\nu$ is the distribution of the stochastic path (Boué & Dupuis, 1998) of $\mathcal{D}$ along timestamps $T$ to $T + T^*$, $z_{M+1:M+L}$ and $y_{M+1:M+L}$ are short for $z$ and $y$ at the timestamps $\{t_{M+1}, \ldots, t_{M+L}\}$, $R_\nu$ is the risk of a learning model $h_\theta$ parameterized by parameters $\theta$.

## 4 Methods

With the prior knowledge of SDEs and EDG, we will now formally present our SDE-EDG approach: To build IFGET (section 4.1), we search sample-to-sample correspondence, which aids in generation of continuous interpolations; Neural SDEs models the trajectories of latent representations (Section 4.2); we construct IFGET and employ it as a regularization mechanism to promote the learning of evolving representations while avoiding the acquisition of invariant representations (Section 4.3).

## 4.1 CONSTRUCT INFINITELY FINED-GRID EVOLVING TRAJECTORY

In EDG, datasets have a considerably small size of domains/timestamps (e.g., at most 30 domains) (Yao et al., 2022). In contrast, models are trained on the historical data for time-series forecasting tasks spanning over at least hundreds of timestamps to predict future states (Addison Howard inversion, 2020; Yu et al., 2016).

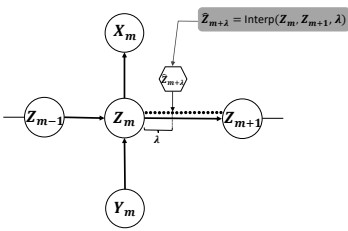

In light of this obstacle, we generate *intermediate domains* by applying interpolations between two consecutive domains. To ensure such interpolations reflect the evolving pattern of the underlying trajectory over domains, one should have the complete trajectory of each individual sample across domains. For example, in weather forecasting, one must have the historical meteorological data of each individual observation station to characterize the climate change trends. Unfortunately, such trajectories usually do not exist in EDG as there is no sample-to-sample correspondence across domains (e.g., we may not have images of the same person at different age stages), thereby preventing the model from tracking the complete trajectories and extracting the evolving trends. To address this issue, we propose to identify sample correspondence between timestamps, which is critical to better alignment of the data structure across domains (Lu et al., 2023; Chen et al., 2022; Blitzer et al., 2006; Das & Lee, 2018).

Figure 1: The graphical model depicts the data generation process of SDE-EDG, where we hide the label superscript $k = Y_m$ and the sample index $i$. $X_m$ is the input sample (for example, images) at $t_m$, $Y_m$ is the label at $t_m$, and $Z_m$ is the latent representation at $t_m$. The continuous-interpolation $\hat{Z}_{m+\lambda}$ is interpolated by the sample pair from IFGET.

Specifically, for each class $k$, we take the closest sample at $t_{m+1}$ to the datapoint $z_m^k$ at $t_m$, as its subsequent state at time $t_{m+1}$ to build sample-to-sample correspondence:

$$\tilde{z}_{i|m+1}^k = \underset{z_{j|m+1}^k \in \mathbb{S}_{m+1}^k}{\arg\min} \ \text{Dist}(z_{i|m}^k, z_{j|m+1}^k), \tag{4}$$

where $\text{Dist} : \mathcal{Z} \times \mathcal{Z} \to [0, +\infty)$ is a distance metric defined over the embedding space, $\mathbb{S}_{m+1}^k$ be the set of $N_B$ data points sampled from $\mathcal{D}_{m+1}$ (short for $\mathcal{D}(z, y|t_{m+1})$) with class $y = k \in \{1, ..., K\}$ in a training iteration. The rationale here lies in the decomposition of latent variables into class-dependent and domain-dependent evolving components (Qin et al., 2022), resulting in samples from the same class in the $m$-th and $(m+1)$-th domains exhibiting smaller distances due to such sample pair's shared class-dependent similarities, while the evolving difference maintains consistent magnitude. Utilizing sample-to-sample correspondence, we gather discrete samples within IFGET. To render it continuous, we generate continuously-interpolated samples bridging the temporal gaps.

With the sample correspondence, we leverage the interpolation function to generate continuous-interpolated samples, such that the interpolation is generated along the approximated individual trajectory of a data point as shown in Figure 1

$$\hat{z}_{i|m+\lambda}^k = \text{Interp}(z_{i|m}^k, \tilde{z}_{i|m+1}^k, \lambda) = (1 - \lambda)z_{i|m}^k + \lambda\tilde{z}_{i|m+1}^k, \forall z_{i|m}^k \in \mathbb{S}_m^k \tag{5}$$

where the interpolation rate $\lambda \in (0, 1)$ is sampled from a Beta distribution $\mathcal{B}(\beta_1, \beta_2)$, $\beta_1$ and $\beta_2$ are the parameters of the Beta distribution, and $\mathbb{S}_m^k$ consists of instances sampled from $k$-th class of $m$-th domain. Here we apply a linear interpolation (Yan et al., 2020; Zhang et al., 2018) as the interpolation function. The continuous-interpolated samples bridge temporal gaps in the discrete evolving trajectory, converting it into an infinitely fine-grained trajectory due to $\lambda \in (0, 1)$. Specifically, as $\lambda$ can be any value between $(0, 1)$, it enables us to approximate time moments between the $m$-th and $(m + 1)$-th timestamps. We theoretically show that the sample complexity of EDG reduces with a smaller temporal interval in Theorem D.3, which leads to a reduction in error. We take interpolations as approximations of samples between time intervals, leading to a smaller time interval and thus a smaller sample complexity. Above all, we construct the Infinitely Fined-grid Evolving Trajectory $\{z_{i|m}^k, \hat{z}_{i|m+\lambda}^k, \tilde{z}_{i|m+1}^k\}_{m=1}^{M-1}$ by leveraging the sample-to-sample correspondence, and collecting the interpolations.

## 4.2 MODELING EDG WITH STOCHASTIC DIFFERENTIAL EQUATIONS

The continuous trajectory in Section 4.1 significantly enhances the capability to capture evolving patterns, but existing EDG methods can not handle the continuous timestamp data. Hence, we

propose to model the data of EDG in the representation space with neural SDEs, since Neural SDEs naturally model continuous temporal trajectories. In contrast, traditional temporal models such as LSTM (Hochreiter & Schmidhuber, 1997) and Markov models (Bishop & Nasrabadi, 2006) are only able to model discrete timestamps.

Here, SDE-EDG learns the temporal dynamics governing the semantic conditional distributions $\mathcal{D}(z|y,t)$ over time. Specifically, SDE-EDG models the temporal trajectory of the data point from the domain at $t_m$ to the arbitrary future timestamp $t_{m'} : t_{m'} > t_m$ of each class $k \in \{1, \ldots, K\}$:

$$\hat{z}_{m'}^k = z_m^k + \int_{t_m}^{t_{m'}} f_k(\hat{z}_s^k, s)ds + \int_{t_m}^{t_{m'}} g_k(\hat{z}_s^k, s)dB_s, \tag{6}$$

where the latent variable $\hat{z}_{m'}^k$ is transformed from $m$-th domains latent variable $z_m^k$, and $f_k$ is the drift function of the $k$-th class to capture the evolving patterns, and $g_k$ is the diffusion function of the $k$-th class to characterize the stochastics of the latent representations. Note that $z$ is the latent variable (representation) induced by $z = \phi(x)$, but $\hat{z}$ is the *synthetic* feature generated by Eq. (6). Hence, SDE-EDG can generate the latent continuous trajectory by gradually transforming the sample representation from the current timestamp $m$ to any desired future timestamp $m'$. Thereby, our latent trajectories of SDE-EDG could effectively align with the collected continuous trajectories IFGET, which prevents overfitting to sparse timestamps.

We design two objective functions to learn the drift functions $f = \{f_k\}_{k=1}^K$ and diffusion functions $g = \{g_k\}_{k=1}^K$ characterized by neural networks: one is aimed to impose Path Alignment Loss in Eq. (7), and another one is downstream classification loss in Eq. (10). By jointly optimizing $\{\phi, f, g\}$ w.r.t these two losses, our approach achieves improved performance on EDG.

### 4.3 ALIGN SDE-EDG WITH IFGET VIA MAXIMUM LIKELIHOOD

Neural SDEs are designed to capture the dynamics and evolution of data over time, particularly in continuous spaces. To fit the SDE-EDG into the evolving stochastic path given observations, we propose the path alignment regularizer by maximizing its likelihood of the IFGET $\{z_{i|m}^k, \hat{z}_{i|m+\lambda}^k, \tilde{z}_{i|m+1}^k\}_{m=1}^{M-1}$:

$$\mathcal{J}_{mle} = \sum_{m=1}^M \sum_{k=1}^K \sum_{i=1}^{N_B} -\frac{1}{MKN_B} \Big( \log \mathcal{D}(z = \tilde{z}_{i|m+1}^k | z = z_{i|m}^k) + \log \mathcal{D}(z = \hat{z}_{i|m+\lambda}^k | z = z_{i|m}^k) \Big), \tag{7}$$

Filling the gap between domains with continuous-interpolated samples results in a continuous and smooth evolving trajectory over time. Taking $\mathcal{J}_{mle}$ as a regularizer brings two advantages to EDG model training: 1) Empirically, the training process of neural SDEs converges faster with $\mathcal{J}_{mle}$, as shown in Figure 4a. 2) $\mathcal{J}_{mle}$ regularizes the latent space to capture evolving patterns. This contributes to learning the evolving patterns in the EDG problem and improves the generalization capability

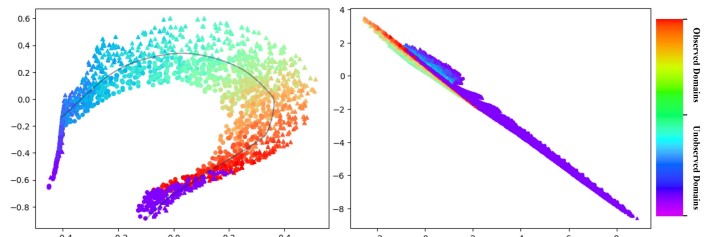

Figure 2: The left and right images depict representations acquired for the Circle dataset through the SDE-EDG and IRM by $\phi$. Distinct classes are distinguished by different shapes (triangles and circles), while various domains are denoted by different colors as indicated by the rainbow bar. SDE-EDG successfully learns representations with a discernible decision boundary, whereas IRM collapses towards a single direction, failing to depict a clear decision boundary.

to target domains as shown in Figure 2. On the other hand, in the absence of $\mathcal{J}_{mle}$, the model learns invariant representations across domains, leading to the occurrence of the Neural Collapse phenomenon (Han et al., 2022), where the latent representations of the same class across the domain collapse to a single point. Consequently, no evolving patterns manifest in the latent representation space as shown in Figure 3.

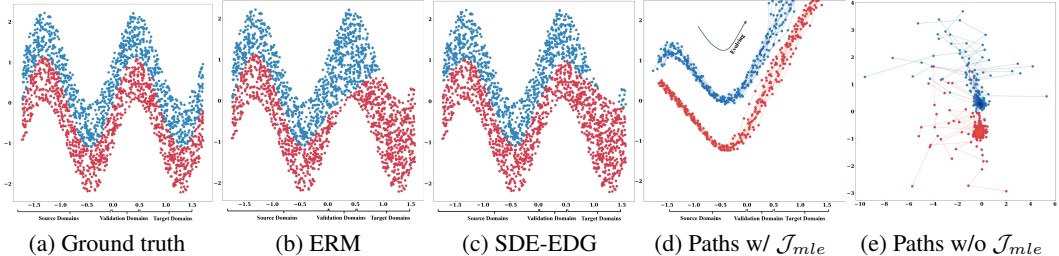

| (a) Ground truth | (b) ERM | (c) SDE-EDG | (d) Paths w/ $\mathcal{J}_{mle}$ | (e) Paths w/o $\mathcal{J}_{mle}$ |

Figure 3: (a) Ground truth of the Sine dataset, and positive and negative labels are red and blue dots separately. (b-c) show prediction results to the Sine made by ERM, and SDE-EDG respectively, positive and negative predictions are red and blue dots separately. (d) Visualized learned evolving paths (synthetic latent variables) **with** the Path Alignment loss $\mathcal{J}_{mle}$ by SDE-EDG learned from the Sine dataset. (e) Synthetic latent variables **without** the Path Alignment loss $\mathcal{J}_{mle}$ by SDE-EDG learned from the Sine dataset. With $\mathcal{J}_{mle}$, the latent evolving dynamics can be correctly characterized, and SDE-EDG can capture the evolving patterns.

### 4.4 SDE-EDG FOR THE PREDICTION LOSS

In this section, we formulate our approach for handling downstream classification tasks. With the Bayes rule, the predictive distribution is

$$\mathcal{D}(y = k|z, t = t_m) = \frac{\mathcal{D}(z|y = k, t = t_m) \times \mathcal{D}(y = k|t = t_m)}{\sum_{k'=1}^{K} \mathcal{D}(z|y = k', t = t_m) \times \mathcal{D}(y = k'|t = t_m)}, \tag{8}$$

where we model $\mathcal{D}(z|y = k, t = t_m)$ with non-parametric model, and $\mathcal{D}(y|t = t_m)$ as a neural net with input as timestamp $t$, function denoted as $r(t)$. In each iteration, we first compute label distribution with respect to time $\mathcal{D}(y|t = t_m) = [\mathcal{D}(y = 1|t = t_m), \ldots, \mathcal{D}(y = K|t = t_m)]$. Specifically, $\mathcal{D}(y = k|t = t_m) = \frac{|\mathbb{S}_m^k|}{\sum_{k'=1}^{K} |\mathbb{S}_m^{k'}|}$, where $\mathbb{S}_m^k$ consists of instances sampled from $k$-th class of $m$-th domain, $|\cdot|$ denotes the size of the set. $r$ is optimized by minimizing $||\frac{|\mathbb{S}_m^k|}{\sum_{k'=1}^{K} |\mathbb{S}_m^{k'}|} - r(t_m)||$.

The conditional distribution $\mathcal{D}(z|y, t)$ modeled by SDEs lacks an analytic expression, and here we approximate it with a non-parametric model. Given that distributions characterized by SDEs may exhibit either uni-modal or multi-modal patterns, it's also advantageous to model multi-modal representations with Neural SDEs (Min et al., 2023). In this context, we present the multi-modal classification loss here but leave the uni-modal loss in the appendix A due to space limitation. To preserve the multi-modal pattern of the latent variables, we employ the non-parametric distribution density method Parzen Window (Parzen, 1962)

$$\mathcal{D}(z|y = k, t = t_m) = \frac{\sum_{\hat{z}_i \in \hat{\mathbb{S}}_m^k} -\exp(-\text{Dist}(z, \hat{z}_i))}{|\hat{\mathbb{S}}_m^k|} \tag{9}$$

where $\hat{\mathbb{S}}_m^k$ includes instances sample from learned SDE-EDG belong to $k$-th class of $m$-th domain. By incorporating the estimations of label distribution ($\mathcal{D}(y|t)$) and conditional distribution ($\mathcal{D}(z|y, t)$), our predictions encompass the temporal evolution of both $\mathcal{D}(z|y, t)$ and $\mathcal{D}(y|t)$.

Model optimization proceeds by minimizing the negative log probability:

$$\mathcal{J}_{cls} = \sum_{m=1}^{M} \sum_{k=1}^{K} \sum_{i=1}^{N_B} -\frac{1}{MKN_B} \log \mathcal{D}(y = k|z = z_i, t = t_m) \tag{10}$$

The ultimate objective function is $\mathcal{J} = \mathcal{J}_{cls} + \alpha \mathcal{J}_{mle}$ The Maximum Likelihood Loss $\mathcal{J}_{mle}$ is a path alignment regularizer that aims to fit the stochastic evolving paths. The hyper-parameter $\alpha > 0$ is the weighting of $\mathcal{J}_{mle}$ to adjust its contribution to the overall loss.

## 5 EXPERIMENTS

To evaluate the effectiveness of SDE-EDG, we verify our method on various datasets, Rotated Gaussian, Sine, and Circle datasets, Rotated MNIST, Portraits, Caltran, Power Supply, and Ocular

---

**Algorithm 1** SDE-EDG (an iteration during the training phase)

---

1: **Input:** $\{S_1, S_2, ..., S_M\}$: $M$ data sets from consecutive domains. $N_B$: the number of instances sampled for each class in an iteration. RANDOMSAMPLE$(S, N)$: a set of $N$ instances sampled uniformly from the set $S$ without replacement. $\mathcal{J} \leftarrow 0$
2: **for** $m$ in $\{1, ..., M-1\}$ **do**
3:     **for** $k$ in $\{1, ..., K\}$ **do**
4:         $\mathbb{S}_m^k \leftarrow$ RANDOMSAMPLE$(S_m^k, N_B)$
5:         $\hat{z}_{m+1}^k = z_m^k + \int_{t_m}^{t_{m+1}} f_k(\hat{z}_s^k, s)ds + \int_{t_m}^{t_{m+1}} g_k(\hat{z}_s^k, s)dB_s$ is computed for $z_m^k \in S^k$
6:         $\mathbb{S}_{m+1}^k \leftarrow$ RANDOMSAMPLE$(S_{m+1}^k, N_B)$
7:         Use Eq. (4) to search subsequent state $\tilde{z}_{m+1}^k$.
8:         Use Eq. (5) to generate continuous-interpolated samples $\hat{z}_{m+\lambda}$.
9:         $\mathcal{J} \leftarrow \mathcal{J} + \alpha \cdot \mathcal{J}_{mle}$, where $\mathcal{J}_{mle}$ is calculated w.r.t Eq. (7).
10:         **for** $(z_i, y_i)$ in $\mathbb{S}_{m+1}^k$ **do**
11:             $\mathcal{J} \leftarrow \mathcal{J} + \frac{1}{MKN_B} \log \mathcal{D}(y = y_i | z = z_i, t = t_m)$
12: Optimize the loss w.r.t. $\mathcal{J}$

---

Disease. The objective of this section aims to answer the following key questions (1) What are the evolving trajectories that SDE-EDG learns, and what is the nature of its learning process? (Shown in Figure 3) (2) How does SDE-EDG compare to other methods in improving EDG performance? (such as Table 1, Figure 5, and Figure 4 (a)-(b)) (3) What is the influence of the Maximum Likelihood Loss on the performance of SDE-EDG? (In Figure 3 (d-e) and Figure 4 (c)-(d))

## 5.1 EXPERIMENTAL SETUP

**Dataset: Rotated Gaussian** (Zeng et al., 2023) consists of 30 domains where each domain has 500 instances generated by the same Gaussian distribution, but the decision boundary rotates from $0°$ to $338°$ with an interval of $12°$. We split the domains into source domains (1-22 domains), intermediate domains (22-25 domains), and target domains (26-30 domains). The intermediate domains are utilized as the validation set. **Circle** (Pesaranghader & Viktor, 2016) contains evolving 30 domains where the instance are sampled from 30 2D Gaussian distributions. The label is assigned using a half-circle curve as the decision boundary (15 source domains, 5 validation domains, and 10 target domains). **Sine** (Pesaranghader & Viktor, 2016) The label is assigned using a sine curve as the decision boundary. We rearrange this dataset by extending it to 24 evolving domains. (12 source domains, 4 validation domains, and 8 target domains) **Rotated MNIST (RMNIST)** (RMNIST) (Ghifary et al., 2015) is composed of MNIST digits of various rotations. We follow (Qin et al., 2022) and extend it to 19 evolving domains via applying the rotations with degree of $\{0°, 15°, 30°, ..., 180°\}$ in order (10 source domains, 3 validation domains, and 6 target domains). **Portraits** (Ginosar et al., 2015) (Yearbook (Yao et al., 2022)) is a real-world dataset that comprises photos of American high school seniors collected over 108 years (1905-2013) for gender classifications. The dataset is divided into 34 domains (19 source domains, 5 validation domains, and 10 target domains). **Caltran** (Hoffman et al., 2014) consists of real-world images captured by a fixed traffic camera deployed in an intersection over time. We divide it into 34 domains by time. The task of Caltran is to classify scenes to identify the presence of one or more vehicles in or approaching the intersection (19 source domains, 5 validation domains, and 10 target domains). **PowerSupply** (Dau et al., 2019) is created for the purpose of predicting the current power supply based on hourly records from an Italian electricity company. It includes 30 domains based on days and each data point is labeled as either morning or afternoon (15 source domains, 5 validation domains, and 10 target domains). **Ocular Disease** (Kaggle, 2020) Ocular Disease Intelligent Recognition (ODIR) is set with three classes: Normal, Diabetes and other diseases. Following the EDG setup, we sort the photographs in ascending order of the age of the patients (27 source domains, 2 validation domains, and 4 target domains).

**Baselines** We compare with following baselines: (1) ERM (Vapnik, 1999); (2) Mixup (Yan et al., 2020); (3) MMD (Li et al., 2018b); (4) MLDG (Li et al., 2018a); (5) IRM (Arjovsky et al., 2019); (6) RSC (Huang et al., 2020); (7) MTL (Blanchard et al., 2021); (8) Fish (Shi et al., 2021); (9) CORAL (Sun & Saenko, 2016); (10) AndMask (Parascandolo et al., 2020); (11) DIVA (Ilse et al., 2020); (12) LSSAE (Qin et al., 2022); (13) GI (Nasery et al., 2021a) (14) DDA (Zeng et al., 2023) (15) DRAIN

Table 1: The comparison of the classification accuracy (%) between SDE-EDG and other baseline methods across the synthetic and real-world datasets. The reported results are the average accuracy of the multiple target domains. ("RG" for RotatedGaussian, "Cir" for Circle, "RM" for Rotated MNIST, "Por" for Portraits, "Cal" for Caltran, "PS" for PowerSupply, "OD" for OcularDisease. GI fails to complete OD due to high time complexity.)

| ALGORITHM | | RG | CIR | SINE | RM | POR | CAL | PS | OD | AVG |
|---|---|---|---|---|---|---|---|---|---|---|
| DG METHODS | ERM | 59.0 | 49.9 | 63.0 | 43.6 | 87.8 | 66.3 | 71.0 | 57.9 | 62.3 |
| | MIXUP | 55.4 | 48.4 | 62.9 | 44.9 | 87.8 | 66.0 | 70.8 | 59.7 | 62.0 |
| | MMD | 56.0 | 50.7 | 55.8 | 44.8 | 87.3 | 57.1 | 70.9 | 57.6 | 60.0 |
| | MLDG | 59.9 | 50.8 | 63.2 | 43.1 | 88.5 | 66.2 | 70.8 | 43.9 | 60.8 |
| | IRM | 47.5 | 51.3 | 63.2 | 39.0 | 85.4 | 64.1 | 70.8 | 46.2 | 58.4 |
| | RSC | 32.8 | 48.0 | 61.5 | 41.7 | 87.3 | 67.0 | 70.9 | 54.5 | 58.0 |
| | MTL | 59.0 | 51.2 | 62.9 | 41.7 | 89.0 | 68.2 | 70.7 | 59.7 | 62.8 |
| | FISH | 41.6 | 48.8 | 62.3 | 44.2 | 88.8 | 68.6 | 70.8 | 48.2 | 59.2 |
| | CORAL | 53.0 | 53.9 | 51.6 | 44.5 | 87.4 | 65.7 | 71.0 | 60.1 | 60.9 |
| | ANDMASK | 76.3 | 47.9 | 69.3 | 42.8 | 70.3 | 56.9 | 70.7 | 51.2 | 60.7 |
| | DIVA | 56.6 | 67.9 | 52.9 | 42.7 | 88.2 | 69.2 | 70.8 | 53.1 | 62.7 |
| EDG METHODS | LSSAE | 48.7 | 73.8 | 71.4 | 46.4 | 89.1 | 70.6 | 71.1 | 52.3 | 65.4 |
| | GI | 50.8 | 54.4 | 65.2 | 44.6 | 88.1 | 70.7 | 71.4 | - | - |
| | DDA | 66.8 | 51.2 | 66.6 | 45.1 | 87.9 | 66.1 | 70.9 | 55.8 | 63.8 |
| | DRAIN | 61.0 | 50.7 | 71.3 | 43.8 | 89.4 | 69.0 | 71.0 | 58.7 | 64.4 |
| | SDE-EDG | **97.7** | **81.5** | **72.2** | **52.6** | **89.6** | **71.3** | **75.7** | **62.6** | **75.4** |

(Bai et al., 2023). All experimental implementations are conducted using the PyTorch packages and are based on the DomainBed (Gulrajani & Lopez-Paz, 2020). To ensure a fair comparison, the neural network architecture (shown in Appendix C.2) of the encoding and classification parts are kept constant across all baselines used in different benchmarks. Five independent experiments with different random seeds are repeated to reduce the variances.

## 5.2 EXPERIMENTAL RESULTS

**SDE-EDG Aligns with the Evolving Trajectories** To find out what SDE-EDG learns, we visualize the temporal trajectories sampled by SDE-EDG in Fig 3d. It should be noted that SDE-EDG learns the evolving dynamics in the latent space $\mathcal{Z}$, which is not directly interpretable. To address this issue, we use an Identity function as the encoding function, which enables us to learn the dynamics directly in raw data space $\mathcal{X}$. Learning in original data space is more challenging but provides us with a more intuitive understanding of the learned dynamics.

Fig 3 shows that the source domains data are between the range $[-\frac{\pi}{2}, 0]$, which means we only train machine learning models with the half sine. The trajectories of the same class are compact within this range, and the boundaries between them have a large margin to ensure good performance. However, the trajectories become looser and the margins become smaller as we move into unobserved domains with timings in the range $(0, \frac{\pi}{2}]$. With a much longer time gap from the source domains, SDE-EDG will eventually fail due to a larger discrepancy between learned paths and the ground truth.

**Quantitative Results** The experimental results of SDE-EDG and other baselines are presented in Table 1, which shows the accuracy the average accuracy for all the target domains (complete results for each domain shown in Table 3-10 for space limitations). SDE-EDG surpasses other baselines' average accuracy on all datasets. The results indicate a significant improvement over the compared traditional DG methods, which is a reasonable finding since traditional DG methods do not address evolving patterns in EDG. Furthermore, SDE-EDG outperforms LSSAE, DDA, and DRAIN (most recently EDG method) by 10.0%, 11.6%, and 11.0% on overall average accuracy, demonstrating the superior ability of our method to capture evolving patterns. The significant accuracy improvements observed in Table 1 indicate consistent enhancements in the performance of EDG tasks. In addition, EDG methods perform consistently better than DG methods, showing the importance of modeling evolving patterns to improve prediction performance in EDG tasks. In particular, the OcularDisease dataset is a challenging medical image classification task, significantly more complex than standard image classification. SDE-EDG demonstrates its ability to capture evolving patterns in demanding real-world scenarios.

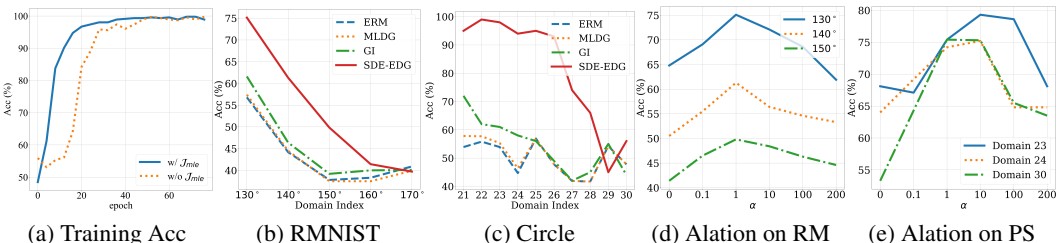

| (a) Training Acc | (b) RMNIST | (c) Circle | (d) Alation on RM | (e) Alation on PS |

Figure 4: (a) Training accuracy convergence trajectories on Portraits (b)-(c) Accuracy with the domain index of RMNIST and Circle Datasets, respectively. (d)-(e) Effects of weighting $\alpha$ on the Maximum Likelihood Loss on RMNIST (RM) and PowerSupply (PS).

Table 2: Rotated MNIST with different temporal gaps $\Delta t$ ($t$ here represents domain index).

| $\Delta t$ \ $t$ | 130° | 140° | 150° | 160° | 170° | 180° | AVG |
|---|---|---|---|---|---|---|---|
| $\Delta t/2$ | $75.6 \pm 0.8$ | $61.8 \pm 0.8$ | $49.9 \pm 0.8$ | $50.0 \pm 0.9$ | $45.1 \pm 0.7$ | $44.1 \pm 0.9$ | 54.4 |
| $\Delta t$ | $75.1 \pm 0.8$ | $61.3 \pm 0.9$ | $49.8 \pm 0.8$ | $49.8 \pm 0.8$ | $39.7 \pm 0.7$ | $39.7 \pm 0.9$ | 52.6 |
| $\Delta t \times 2$ | $58.6 \pm 0.8$ | $49.1 \pm 0.7$ | $45.6 \pm 0.7$ | $42.4 \pm 0.8$ | $36.9 \pm 0.8$ | $36.1 \pm 0.8$ | 44.8 |

Figure 4b and figure 4c plot the accuracy trajectory of the baselines (ERM, MLDG, GI) and SDE-EDG across domains for RMNIST and Circle datasets, which show the superiority of SDE-EDG on the other 3 baselines with large margin improvements. SDE-EDG could keep large improvements initially, but eventually, as the distance between the target and source domains increases, all methods will achieve similar performance. Therefore, we conclude, that in EDG, models can achieve generalization in the relatively near future.

**Ablations: the Impact of Maximum Likelihood Loss on EDG Classifications**   We conducted an ablation study on the RMNIST and PowerSupply datasets to evaluate the effectiveness of the proposed Maximum Likelihood Loss $\mathcal{J}_{mle}$, which aims to train SDE-EDG to fit IFGET. Figure 4d and Figure 4e show that SDE-EDG achieves the best performance with $\alpha = 1$ on the RMNIST dataset and $\alpha = 10$ on the PowerSupply dataset. The above empirical results suggest that $\mathcal{J}_{mle}$ improves the performance of EDG by aligning the underlying evolving paths (optimizing the Path Alignment loss $\mathcal{J}_{pa}$) and quantifying stochastic uncertainties (minimize Stochastic Uncertainty loss $\mathcal{J}_{su}$), details proof in Appendix D.1. Specifically, when we only apply classification loss with $\alpha = 0$, the performance is the worst. On the other hand, with a much larger $\alpha = 200$, SDE-EDG focuses on aligning the evolving paths and giving lower importance to classification tasks. Thus, aligning stochastic evolving processes improves performance in EDG.

**Ablations: $\Delta t$ influence on EDG performance**   In Table 2, we set interval $\Delta t$ to 5°, and 20° between source domains with SDE-EDG, where the 10° interval is in the original setting. With smaller $\Delta t$ ($\Delta t \times 2 \rightarrow \Delta t \rightarrow \Delta t/2$), the accuracy experiences a consistent improvement, a finding that aligns with our motivation: a smaller temporal gap between domains reduces the generalization error. Therefore, interpolations between temporal gaps as an approximation to the sample at an arbitrary timestamp would lead to reduced $\Delta t$ and overcoming overfitting to available limited timestamps.

## 6   CONCLUSION

This work presents a new approach SDE-EDG for modeling Evolving Domain Generalization (EDG). Our approach involves constructing IFGET by identifying sample-to-sample correspondence and generating continuous-interpolated samples via linear interpolations. Subsequently, we employ Stochastic Differential Equations (SDE) and train it in alignment with IFGET. Our contribution lies in revealing the importance of capturing the evolving patterns through the collected individual's temporal trajectories, and of interpolating between time intervals to mitigate the issue of the limited number of source timestamps, which effectively prevents SDE-EDG from overfitting to the limited timestamps. We also provide a theoretical analysis demonstrating that our method can reduce the generalization risk.

ACKNOWLEDGEMENTS

We appreciate constructive feedback from anonymous reviewers and meta-reviewers. This work is supported by the Natural Sciences and Engineering Research Council of Canada (NSERC), Discovery Grants program.

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

## A  UNI-MODAL CLASSIFICATION LOSS

We calculate the Evolving Centroid of domain at $t_m$ as the mean vector of the instances belonging to $\hat{\mathbb{S}}_m^k$, $\hat{c}_m^k = \frac{1}{N_B} \sum_{(\hat{z}_{i|m}^k, \hat{y}_{i|m}^k) \in \hat{\mathbb{S}}_m^k} \hat{z}_{i|m}^k$, where $\hat{\mathbb{S}}_m^k$ be the set of $N_B$ data points sampled from $\hat{\mathcal{D}}_m$ (short for $\hat{\mathcal{D}}(z, y|t_m)$) in a training iteration by SDE-EDG, $i|m$ stands for the $i$-th instance at time $t_m$, and $N_B$ is the number of instances sampled in a training iteration belonging to $k$-th class of the domain at timestamp $t_m$.

At each timestamp, the Evolving Centroids exhibit holistic evolving patterns reflecting the dynamics of class-conditional distribution. This differs from the conventional Domain Adaptation (DA) or Domain Generalization (DG) methods, which aim to learn the invariant features by maintaining a global centroid for each class(Arjovsky et al., 2019; Li et al., 2018b; Muandet et al., 2013; Nguyen et al., 2021). Instead, we do not aim to align the class centroid across all domains but allow the Evolving Centroids to evolve over time, which offers a more accurate representation of the data distribution's temporal changes. The predictive distribution for $\mathcal{D}_m$ is

$$\mathcal{D}(y = k|z, t = t_m) = \frac{\exp(-\text{Dist}(z, \hat{c}_m^k))}{\sum_{k'=1}^{K} \exp(-\text{Dist}(z, \hat{c}_m^{k'}))}, \tag{11}$$

During the training stage, at each step, we sample $N_B$ samples from each class $k$ in $\hat{\mathcal{S}}_m$, which is used to compute Evolving Centroid $\hat{c}_m^k$ to make predictions for data in $\mathcal{S}_m$.

In the inference stage, we search the closest Evolving Centroids sampled by SDE-EDG at a given timestamp, and assign the class of the closest Evolving Centroid as the classification prediction.

## B  FULL EXPERIMENTAL RESULTS

In this section, we present complete experimental results to validate the efficacy of our proposed evolving domain generalization task. Our findings demonstrate that, across a wide range of scenarios, our approach consistently outperforms existing domain generalization baselines, thus establishing a new state-of-the-art performance on existing benchmarks. These results affirm the effectiveness of our method in addressing the challenges posed by evolving domains. By achieving superior performance, our approach shows its applicability to real-world problems.

Table 3: Rotated Gaussian. We show the results on each target domain by domain index.

| ALGORITHM | 26 | 27 | 28 | 29 | 30 | AVG |
|---|---|---|---|---|---|---|
| ERM | $58.8 \pm 1.3$ | $58.0 \pm 1.5$ | $57.8 \pm 1.3$ | $62.0 \pm 1.1$ | $58.6 \pm 1.1$ | 59.0 |
| MIXUP | $56.2 \pm 1.5$ | $63.4 \pm 3.0$ | $56.8 \pm 1.4$ | $49.4 \pm 1.5$ | $41.4 \pm 2.0$ | 55.4 |
| MMD | $53.8 \pm 0.9$ | $53.0 \pm 1.0$ | $52.8 \pm 1.0$ | $57.0 \pm 1.6$ | $63.6 \pm 1.9$ | 56.0 |
| MLDG | $64.0 \pm 1.6$ | $58.8 \pm 0.9$ | $57.2 \pm 1.0$ | $57.2 \pm 1.1$ | $62.2 \pm 1.3$ | 59.9 |
| IRM | $56.8 \pm 1.9$ | $55.8 \pm 3.1$ | $51.8 \pm 2.3$ | $41.6 \pm 1.6$ | $31.4 \pm 2.1$ | 47.5 |
| RSC | $31.6 \pm 1.9$ | $33.8 \pm 2.2$ | $31.8 \pm 1.6$ | $31.8 \pm 2.0$ | $35.2 \pm 2.0$ | 32.8 |
| MTL | $54.2 \pm 2.2$ | $54.6 \pm 1.3$ | $58.8 \pm 1.4$ | $59.8 \pm 1.6$ | $67.6 \pm 2.2$ | 59.0 |
| FISH | $52.0 \pm 2.4$ | $49.6 \pm 3.4$ | $44.6 \pm 2.0$ | $35.6 \pm 1.4$ | $26.2 \pm 1.7$ | 41.6 |
| CORAL | $54.8 \pm 1.6$ | $54.0 \pm 0.6$ | $53.8 \pm 1.0$ | $52.0 \pm 0.8$ | $50.6 \pm 1.6$ | 53.0 |
| ANDMASK | $81.8 \pm 1.2$ | $79.4 \pm 1.6$ | $75.4 \pm 1.6$ | $75.4 \pm 1.3$ | $69.4 \pm 1.7$ | 76.3 |
| DIVA | $59.0 \pm 1.5$ | $55.8 \pm 0.9$ | $53.6 \pm 0.7$ | $59.2 \pm 1.3$ | $55.6 \pm 1.5$ | 56.6 |
| LSSAE | $50.6 \pm 0.9$ | $50.8 \pm 2.3$ | $43.4 \pm 1.4$ | $48.4 \pm 2.4$ | $50.4 \pm 2.1$ | 48.7 |
| GI | $51.6 \pm 2.9$ | $53.2 \pm 2.2$ | $47.4 \pm 1.8$ | $49.6 \pm 2.7$ | $52.4 \pm 2.0$ | 50.8 |
| DDA | $77.4 \pm 1.3$ | $75.0 \pm 1.5$ | $68.2 \pm 1.1$ | $60.8 \pm 1.0$ | $52.8 \pm 1.5$ | 66.8 |
| DRAIN | $73.2 \pm 2.9$ | $70.0 \pm 1.7$ | $63.8 \pm 2.4$ | $53.2 \pm 2.2$ | $45.0 \pm 1.2$ | 61.0 |
| SDE-EDG (OURS) | $98.4 \pm 0.6$ | $97.0 \pm 0.6$ | $97.4 \pm 0.8$ | $97.8 \pm 0.6$ | $97.8 \pm 0.7$ | 97.7 |

Table 4: Circle. We show the results on each target domain by domain index.

| ALGORITHM | 21 | 22 | 23 | 24 | 25 | 26 | 27 | 28 | 29 | 30 | AVG |
|---|---|---|---|---|---|---|---|---|---|---|---|
| ERM | 53.9 ± 3.5 | 55.8 ± 4.8 | 53.9 ± 5.2 | 44.7 ± 6.3 | 56.9 ± 4.4 | 47.8 ± 5.8 | 41.9 ± 7.5 | 41.7 ± 5.9 | 54.2 ± 3.0 | 47.8 ± 5.7 | 49.9 |
| MIXUP | 48.6 ± 3.8 | 51.7 ± 4.0 | 49.4 ± 4.5 | 43.6 ± 5.8 | 56.9 ± 4.4 | 47.8 ± 5.8 | 41.9 ± 7.5 | 41.7 ± 5.9 | 54.2 ± 3.0 | 47.8 ± 5.7 | 48.4 |
| MMD | 50.0 ± 3.9 | 53.6 ± 4.4 | 55.0 ± 4.3 | 51.9 ± 6.0 | 60.8 ± 3.9 | 49.7 ± 7.2 | 41.9 ± 7.5 | 41.7 ± 5.9 | 54.2 ± 3.0 | 47.8 ± 5.7 | 50.7 |
| MLDG | 57.8 ± 3.6 | 57.7 ± 5.0 | 55.3 ± 4.9 | 46.4 ± 6.8 | 56.9 ± 4.4 | 47.8 ± 5.8 | 41.9 ± 7.5 | 41.7 ± 5.9 | 54.2 ± 3.0 | 47.8 ± 5.7 | 50.8 |
| IRM | 57.8 ± 3.9 | 59.4 ± 5.4 | 56.9 ± 4.9 | 48.1 ± 7.4 | 57.5 ± 4.3 | 47.8 ± 5.8 | 41.9 ± 7.5 | 41.7 ± 5.9 | 54.2 ± 3.0 | 47.8 ± 5.7 | 51.3 |
| RSC | 45.3 ± 3.6 | 51.4 ± 3.8 | 49.4 ± 4.5 | 43.6 ± 5.8 | 56.9 ± 4.4 | 47.8 ± 5.8 | 41.9 ± 7.5 | 41.7 ± 5.9 | 54.2 ± 3.0 | 47.8 ± 5.7 | 48.0 |
| MTL | 61.4 ± 2.2 | 57.2 ± 6.4 | 53.3 ± 5.1 | 48.3 ± 6.2 | 56.9 ± 4.8 | 49.2 ± 3.7 | 43.3 ± 5.0 | 45.8 ± 2.8 | 54.2 ± 5.7 | 42.2 ± 4.9 | 51.2 |
| FISH | 51.7 ± 3.7 | 53.1 ± 3.7 | 49.4 ± 4.5 | 43.6 ± 5.8 | 56.9 ± 4.4 | 47.8 ± 5.8 | 41.9 ± 7.5 | 41.7 ± 5.9 | 54.2 ± 3.0 | 47.8 ± 5.7 | 48.8 |
| CORAL | 65.3 ± 3.2 | 63.9 ± 4.4 | 60.0 ± 4.8 | 56.4 ± 6.0 | 60.2 ± 4.3 | 47.8 ± 5.8 | 41.9 ± 7.5 | 41.7 ± 5.9 | 54.2 ± 3.0 | 47.8 ± 5.7 | 53.9 |
| ANDMASK | 42.8 ± 3.4 | 50.6 ± 4.1 | 49.4 ± 4.5 | 43.6 ± 5.8 | 56.9 ± 4.4 | 47.8 ± 5.8 | 41.9 ± 7.5 | 41.7 ± 5.9 | 54.2 ± 3.0 | 49.7 ± 5.4 | 47.9 |
| DIVA | 81.3 ± 3.5 | 76.3 ± 4.2 | 74.7 ± 4.6 | 56.7 ± 5.1 | 67.0 ± 6.1 | 62.3 ± 5.1 | 62.0 ± 5.6 | 66.3 ± 4.1 | 70.3 ± 5.6 | 62.0 ± 4.2 | 67.9 |
| LSSAE | 95.8 ± 1.9 | 95.6 ± 2.1 | 93.5 ± 2.9 | 96.3 ± 1.8 | 83.8 ± 5.2 | 74.3 ± 3.6 | 51.9 ± 5.6 | 52.3 ± 8.1 | 46.5 ± 9.2 | 48.4 ± 5.3 | 73.8 |
| GI | 72.0 ± 4.9 | 62.0 ± 4.9 | 61.0 ± 4.9 | 58.0 ± 2.8 | 56.0 ± 4.9 | 49.0 ± 0.7 | 42.0 ± 1.4 | 45.0 ± 2.1 | 55.0 ± 3.5 | 44.0 ± 0.0 | 54.4 |
| DDA | 71.0 ± 0.7 | 56.0 ± 0.0 | 51.0 ± 4.9 | 44.0 ± 2.8 | 55.0 ± 0.7 | 49.0 ± 2.1 | 42.0 ± 7.1 | 45.0 ± 0.7 | 55.0 ± 3.5 | 44.0 ± 4.2 | 51.2 |
| DRAIN | 48.0 ± 2.1 | 52.0 ± 0.7 | 54.0 ± 3.5 | 47.0 ± 1.4 | 58.0 ± 3.5 | 52.0 ± 3.5 | 45.0 ± 4.2 | 48.0 ± 4.9 | 58.0 ± 4.9 | 47.0 ± 2.8 | 50.7 |
| SDE-EDG (OURS) | 95.0 ± 0.7 | 99.0 ± 1.4 | 98.0 ± 1.4 | 94.0 ± 1.4 | 95.0 ± 1.4 | 93.0 ± 7.8 | 74.0 ± 5.7 | 66.0 ± 0.7 | 45.0 ± 6.4 | 56.0 ± 4.5 | 81.5 |

Table 5: Sine. We show the results on each target domain by domain index.

| ALGORITHM | 17 | 18 | 19 | 20 | 21 | 22 | 23 | 24 | AVG |
|---|---|---|---|---|---|---|---|---|---|
| ERM | 71.4 ± 6.1 | 91.0 ± 1.5 | 81.6 ± 2.4 | 53.4 ± 2.9 | 51.1 ± 6.7 | 54.3 ± 4.7 | 49.5 ± 4.8 | 51.7 ± 5.0 | 63.0 |
| MIXUP | 63.1 ± 5.9 | 93.5 ± 1.7 | 80.6 ± 3.8 | 52.8 ± 2.9 | 60.3 ± 7.2 | 54.2 ± 2.7 | 49.5 ± 4.4 | 49.3 ± 8.0 | 62.9 |
| MMD | 57.0 ± 4.2 | 57.1 ± 4.1 | 47.6 ± 5.4 | 50.0 ± 1.8 | 55.1 ± 6.7 | 54.4 ± 4.7 | 49.5 ± 4.8 | 51.7 ± 5.0 | 55.8 |
| MLDG | 69.2 ± 4.2 | 67.7 ± 4.1 | 52.1 ± 5.4 | 50.7 ± 1.8 | 51.1 ± 6.7 | 54.3 ± 4.7 | 49.5 ± 4.8 | 51.7 ± 5.0 | 63.2 |
| IRM | 66.9 ± 6.2 | 81.1 ± 3.2 | 88.5 ± 3.0 | 56.6 ± 6.0 | 57.2 ± 5.8 | 53.7 ± 5.1 | 49.5 ± 2.2 | 51.7 ± 5.4 | 63.2 |
| RSC | 61.3 ± 6.6 | 83.5 ± 1.9 | 84.5 ± 2.6 | 52.8 ± 2.8 | 55.1 ± 6.7 | 54.4 ± 4.7 | 49.5 ± 4.8 | 51.7 ± 5.0 | 61.5 |
| MTL | 70.6 ± 6.6 | 91.6 ± 1.2 | 79.9 ± 3.4 | 51.0 ± 4.7 | 60.3 ± 7.6 | 53.6 ± 5.2 | 49.5 ± 5.3 | 46.9 ± 5.9 | 62.9 |
| FISH | 66.1 ± 6.9 | 82.0 ± 2.7 | 87.5 ± 2.4 | 55.2 ± 3.0 | 51.1 ± 6.7 | 54.3 ± 4.7 | 49.5 ± 4.8 | 51.7 ± 5.0 | 62.3 |
| CORAL | 60.0 ± 5.3 | 57.1 ± 4.2 | 48.6 ± 6.4 | 50.7 ± 1.8 | 49.7 ± 6.2 | 48.6 ± 4.6 | 46.3 ± 5.0 | 51.7 ± 5.0 | 51.6 |
| ANDMASK | 44.2 ± 5.1 | 42.9 ± 4.2 | 54.2 ± 7.0 | 71.9 ± 1.9 | 86.4 ± 3.2 | 90.4 ± 2.9 | 88.1 ± 3.4 | 76.4 ± 3.7 | 69.3 |
| DIVA | 79.0 ± 6.6 | 60.8 ± 1.9 | 47.6 ± 2.6 | 50.0 ± 2.8 | 55.1 ± 6.7 | 51.9 ± 4.7 | 38.6 ± 4.8 | 40.4 ± 5.0 | 52.9 |
| LSSAE | 93.0 ± 1.7 | 86.9 ± 0.7 | 69.2 ± 1.5 | 63.8 ± 3.8 | 68.8 ± 2.5 | 76.8 ± 4.8 | 63.9 ± 1.3 | 49.0 ± 3.1 | 71.4 |
| GI | 77.0 ± 0.7 | 84.2 ± 4.4 | 89.1 ± 0.6 | 60.9 ± 3.6 | 55.1 ± 0.8 | 53.5 ± 6.0 | 49.8 ± 4.1 | 52.0 ± 2.8 | 65.2 |
| DDA | 43.0 ± 0.7 | 47.2 ± 0.6 | 74.2 ± 1.6 | 89.6 ± 1.1 | 75.5 ± 1.1 | 70.0 ± 7.1 | 59.1 ± 2.2 | 74.0 ± 1.4 | 66.6 |
| DRAIN | 43.0 ± 4.9 | 44.6 ± 4.7 | 69.3 ± 0.9 | 89.6 ± 1.1 | 82.8 ± 4.8 | 78.5 ± 2.5 | 70.2 ± 4.1 | 92.0 ± 5.7 | 71.3 |
| SDE-EDG (OURS) | 99.0 ± 0.7 | 96.9 ± 0.6 | 90.0 ± 4.2 | 88.8 ± 0.6 | 64.9 ± 2.2 | 52.4 ± 0.3 | 43.0 ± 4.9 | 42.3 ± 4.0 | 72.2 |

Table 6: RMNIST. We show the results on each target domain by domain index.

| ALGORITHM | 130° | 140° | 150° | 160° | 170° | 180° | AVG |
|---|---|---|---|---|---|---|---|
| ERM | 56.8 ± 0.9 | 44.2 ± 0.8 | 37.8 ± 0.6 | 38.3 ± 0.8 | 40.9 ± 0.8 | 43.6 ± 0.8 | 43.6 |
| MIXUP | 61.3 ± 0.7 | 47.4 ± 0.8 | 39.1 ± 0.7 | 38.3 ± 0.7 | 40.5 ± 0.8 | 42.8 ± 0.9 | 44.9 |
| MMD | 59.2 ± 0.9 | 46.0 ± 0.8 | 39.0 ± 0.7 | 39.3 ± 0.8 | 41.6 ± 0.7 | 43.7 ± 0.8 | 44.8 |
| MLDG | 57.4 ± 0.7 | 44.5 ± 0.9 | 37.5 ± 0.8 | 37.5 ± 0.8 | 39.9 ± 0.8 | 42.0 ± 0.9 | 43.1 |
| IRM | 47.7 ± 0.9 | 38.5 ± 0.7 | 34.1 ± 0.7 | 35.7 ± 0.8 | 37.8 ± 0.8 | 40.3 ± 0.8 | 39.0 |
| RSC | 54.1 ± 0.9 | 41.9 ± 0.8 | 35.8 ± 0.7 | 37.0 ± 0.8 | 39.8 ± 0.8 | 41.6 ± 0.8 | 41.7 |
| MTL | 54.8 ± 0.9 | 43.1 ± 0.8 | 36.4 ± 0.8 | 36.1 ± 0.8 | 39.1 ± 0.9 | 40.9 ± 0.8 | 41.7 |
| FISH | 60.8 ± 0.8 | 47.8 ± 0.8 | 39.2 ± 0.8 | 37.6 ± 0.7 | 39.0 ± 0.8 | 40.7 ± 0.7 | 44.2 |
| CORAL | 58.8 ± 0.9 | 46.2 ± 0.8 | 38.9 ± 0.7 | 38.5 ± 0.8 | 41.3 ± 0.8 | 43.5 ± 0.8 | 44.5 |
| ANDMASK | 53.5 ± 0.9 | 42.9 ± 0.8 | 37.8 ± 0.7 | 38.6 ± 0.8 | 40.8 ± 0.8 | 43.2 ± 0.8 | 42.8 |
| DIVA | 58.3 ± 0.8 | 45.0 ± 0.8 | 37.6 ± 0.8 | 36.9 ± 0.7 | 38.1 ± 0.8 | 40.1 ± 0.8 | 42.7 |
| LSSAE | 64.1 ± 0.8 | 51.6 ± 0.8 | 43.4 ± 0.8 | 38.6 ± 0.7 | 40.3 ± 0.8 | 40.4 ± 0.8 | 46.4 |
| GI | 61.6 ± 0.9 | 46.4 ± 0.9 | 39.2 ± 0.8 | 40.0 ± 0.8 | 40.1 ± 0.8 | 40.1 ± 0.7 | 44.6 |
| DDA | 60.7 ± 0.8 | 50.0 ± 0.8 | 42.6 ± 0.8 | 39.6 ± 0.8 | 38.0 ± 0.8 | 39.7 ± 0.8 | 45.1 |
| DRAIN | 59.5 ± 0.8 | 45.4 ± 0.8 | 40.2 ± 0.7 | 37.2 ± 0.7 | 39.6 ± 0.8 | 41.0 ± 0.7 | 43.8 |
| SDE-EDG (OURS) | 75.1 ± 0.8 | 61.3 ± 0.9 | 49.8 ± 0.8 | 49.8 ± 0.8 | 39.7 ± 0.7 | 39.7 ± 0.9 | 52.6 |

Table 7: Portraits. We show the results on each target domain by domain index.

| ALGORITHM | 25 | 26 | 27 | 28 | 29 | 30 | 31 | 32 | 33 | 34 | AVG |
|---|---|---|---|---|---|---|---|---|---|---|---|
| ERM | 75.5 ± 0.9 | 83.8 ± 0.9 | 88.5 ± 0.8 | 93.3 ± 0.7 | 93.4 ± 0.6 | 92.1 ± 0.7 | 90.6 ± 0.8 | 84.3 ± 0.9 | 88.5 ± 0.9 | 87.9 ± 1.4 | 87.8 |
| MIXUP | 75.5 ± 0.9 | 83.8 ± 0.9 | 88.5 ± 0.8 | 93.3 ± 0.7 | 93.4 ± 0.6 | 92.1 ± 0.7 | 90.6 ± 0.8 | 84.3 ± 0.9 | 88.5 ± 0.9 | 87.9 ± 1.4 | 87.8 |
| MMD | 74.0 ± 1.0 | 83.8 ± 0.8 | 87.2 ± 0.8 | 93.0 ± 0.7 | 93.0 ± 0.6 | 91.9 ± 0.7 | 90.9 ± 0.7 | 84.7 ± 1.4 | 88.3 ± 0.9 | 85.8 ± 1.8 | 87.3 |
| MLDG | 76.4 ± 0.8 | 85.5 ± 0.9 | 90.1 ± 0.7 | 94.3 ± 0.6 | 93.5 ± 0.6 | 92.0 ± 0.7 | 90.8 ± 0.8 | 85.6 ± 1.1 | 89.3 ± 0.8 | 87.6 ± 1.6 | 88.5 |
| IRM | 74.2 ± 0.9 | 83.5 ± 0.9 | 88.5 ± 0.8 | 91.0 ± 0.8 | 90.4 ± 0.7 | 87.3 ± 0.8 | 87.0 ± 0.9 | 80.4 ± 1.5 | 86.7 ± 0.9 | 85.1 ± 1.8 | 85.4 |
| RSC | 75.2 ± 0.9 | 84.7 ± 0.8 | 87.9 ± 0.7 | 93.3 ± 0.7 | 92.5 ± 0.7 | 91.0 ± 0.7 | 90.0 ± 0.7 | 84.6 ± 1.2 | 88.2 ± 0.8 | 85.8 ± 1.9 | 87.3 |
| MTL | 78.2 ± 0.9 | 86.5 ± 0.8 | 90.9 ± 0.8 | 94.2 ± 0.7 | 93.8 ± 0.6 | 92.0 ± 0.7 | 91.2 ± 0.7 | 86.0 ± 1.2 | 89.3 ± 0.8 | 87.4 ± 1.4 | 89.0 |
| FISH | 78.6 ± 0.9 | 86.9 ± 0.8 | 89.5 ± 0.8 | 93.5 ± 0.7 | 93.3 ± 0.6 | 92.1 ± 0.6 | 91.1 ± 0.7 | 86.2 ± 1.3 | 88.7 ± 0.9 | 87.7 ± 1.6 | 88.8 |
| CORAL | 74.6 ± 0.9 | 84.6 ± 0.8 | 87.9 ± 0.8 | 93.3 ± 0.6 | 92.7 ± 0.7 | 91.5 ± 0.7 | 90.7 ± 0.7 | 84.6 ± 1.5 | 88.1 ± 0.9 | 85.9 ± 1.9 | 87.4 |
| ANDMASK | 62.0 ± 1.1 | 70.8 ± 1.1 | 67.0 ± 1.2 | 70.2 ± 1.1 | 75.2 ± 1.1 | 74.1 ± 1.0 | 72.7 ± 1.1 | 64.7 ± 1.6 | 77.3 ± 1.1 | 74.9 ± 2.1 | 70.9 |
| DIVA | 76.2 ± 1.0 | 86.6 ± 0.8 | 88.8 ± 0.8 | 93.5 ± 0.7 | 93.1 ± 0.6 | 91.6 ± 0.6 | 91.1 ± 0.7 | 84.7 ± 1.3 | 89.1 ± 0.8 | 87.0 ± 1.5 | 88.2 |
| LSSAE | 77.7 ± 0.9 | 87.1 ± 0.8 | 90.8 ± 0.7 | 94.3 ± 0.6 | 94.3 ± 0.6 | 92.2 ± 0.6 | 91.2 ± 0.7 | 86.7 ± 1.1 | 89.6 ± 0.8 | 86.9 ± 1.4 | 89.1 |
| GI | 77.8 ± 1.2 | 86.6 ± 1.3 | 90.8 ± 1.1 | 95.3 ± 1.3 | 89.3 ± 1.1 | 88.9 ± 1.2 | 84.1 ± 1.8 | 87.7 ± 1.0 | 87.5 ± 2.0 | | 88.1 |
| DDA | 76.0 ± 1.0 | 85.6 ± 0.8 | 88.6 ± 0.8 | 93.6 ± 0.6 | 92.9 ± 0.7 | 92.9 ± 0.6 | 90.3 ± 0.8 | 84.3 ± 1.2 | 88.7 ± 0.8 | 85.9 ± 1.2 | 87.9 |
| DRAIN | 77.7 ± 0.8 | 86.2 ± 0.8 | 90.6 ± 0.6 | 94.8 ± 0.5 | 94.4 ± 0.6 | 92.8 ± 0.7 | 92.2 ± 0.6 | 87.2 ± 1.2 | 89.9 ± 0.8 | 87.9 ± 1.1 | 89.4 |
| SDE-EDG (OURS) | 78.6 ± 0.8 | 86.6 ± 0.9 | 90.1 ± 0.8 | 94.8 ± 0.6 | 94.5 ± 0.6 | 93.3 ± 0.7 | 92.1 ± 0.7 | 87.9 ± 1.3 | 89.6 ± 0.9 | 89.0 ± 1.1 | 89.6 |

Table 8: Caltran. We show the results on each target domain by domain index.

| ALGORITHM | 25 | 26 | 27 | 28 | 29 | 30 | 31 | 32 | 33 | 34 | AVG |
|---|---|---|---|---|---|---|---|---|---|---|---|
| ERM | 29.9 ± 3.5 | 88.4 ± 2.1 | 61.1 ± 3.5 | 56.3 ± 3.2 | 90.0 ± 1.6 | 60.1 ± 2.5 | 55.5 ± 3.5 | 88.8 ± 2.4 | 57.1 ± 3.5 | 50.5 ± 5.2 | 66.3 |
| MIXUP | 53.6 ± 3.9 | 89.0 ± 2.0 | 61.8 ± 2.4 | 55.7 ± 2.9 | 88.2 ± 2.1 | 58.6 ± 3.0 | 52.3 ± 3.7 | 88.6 ± 2.7 | 57.1 ± 3.0 | 55.1 ± 4.3 | 66.0 |
| MMD | 30.2 ± 2.1 | 92.7 ± 1.7 | 56.4 ± 3.7 | 39.1 ± 3.2 | 93.6 ± 1.7 | 52.1 ± 3.2 | 42.8 ± 3.0 | 92.1 ± 2.2 | 42.1 ± 3.8 | 29.4 ± 3.8 | 57.1 |
| MLDG | 54.8 ± 4.1 | 88.6 ± 2.6 | 62.2 ± 3.6 | 55.1 ± 4.1 | 88.3 ± 1.7 | 60.9 ± 4.3 | 51.7 ± 2.6 | 89.0 ± 1.9 | 56.5 ± 3.4 | 55.3 ± 4.8 | 66.2 |
| IRM | 46.4 ± 3.7 | 90.8 ± 1.7 | 60.8 ± 3.4 | 52.9 ± 3.1 | 91.8 ± 1.7 | 56.6 ± 3.1 | 52.1 ± 2.9 | 90.9 ± 2.6 | 55.6 ± 3.9 | 43.1 ± 5.5 | 64.1 |
| RSC | 57.2 ± 3.0 | 88.4 ± 2.6 | 62.6 ± 3.0 | 56.5 ± 3.7 | 88.0 ± 2.4 | 59.4 ± 3.0 | 51.9 ± 2.9 | 90.0 ± 2.0 | 59.4 ± 2.9 | 56.0 ± 3.1 | 67.0 |
| MTL | 64.2 ± 3.0 | 87.2 ± 2.5 | 64.9 ± 3.9 | 60.0 ± 4.8 | 84.5 ± 2.2 | 60.6 ± 3.5 | 52.6 ± 3.7 | 83.9 ± 2.9 | 58.2 ± 4.1 | 65.7 ± 5.6 | 68.2 |
| FISH | 61.1 ± 3.5 | 88.2 ± 1.5 | 64.7 ± 4.0 | 57.9 ± 3.1 | 88.3 ± 2.2 | 59.9 ± 3.0 | 57.5 ± 2.7 | 87.4 ± 2.8 | 57.7 ± 3.7 | 63.0 ± 6.1 | 68.6 |
| CORAL | 50.4 ± 3.0 | 90.8 ± 2.0 | 61.2 ± 3.8 | 55.0 ± 2.5 | 92.0 ± 1.7 | 56.8 ± 3.8 | 52.0 ± 3.8 | 90.9 ± 1.6 | 56.8 ± 2.4 | 50.9 ± 5.6 | 65.7 |
| ANDMASK | 30.0 ± 2.2 | 92.7 ± 1.7 | 56.2 ± 3.8 | 39.1 ± 3.2 | 93.6 ± 1.7 | 51.6 ± 3.2 | 42.6 ± 2.9 | 92.1 ± 2.2 | 41.2 ± 3.7 | 29.9 ± 3.6 | 56.9 |
| DIVA | 60.6 ± 2.9 | 90.1 ± 1.7 | 67.5 ± 3.1 | 58.9 ± 3.5 | 92.9 ± 1.6 | 58.7 ± 3.3 | 53.8 ± 3.6 | 89.8 ± 1.7 | 61.8 ± 4.8 | 62.0 ± 3.4 | 69.2 |
| LSSAE | 63.4 ± 3.4 | 92.1 ± 2.0 | 62.6 ± 4.7 | 58.8 ± 4.4 | 92.9 ± 1.6 | 62.0 ± 3.9 | 54.3 ± 3.0 | 92.1 ± 2.2 | 60.5 ± 3.8 | 67.4 ± 3.6 | 70.6 |
| GI | 68.8 ± 3.1 | 86.6 ± 1.9 | 65.5 ± 3.2 | 60.6 ± 4.3 | 88.8 ± 2.4 | 58.5 ± 3.6 | 53.1 ± 2.8 | 88.7 ± 2.1 | 63.7 ± 2.9 | 73.0 ± 5.1 | 70.7 |
| DDA | 31.0 ± 3.4 | 92.6 ± 1.8 | 56.8 ± 0.9 | 59.0 ± 2.8 | 94.0 ± 2.2 | 61.7 ± 2.3 | 52.9 ± 2.3 | 92.9 ± 2.2 | 57.8 ± 5.3 | 62.9 ± 5.8 | 66.1 |
| DRAIN | 66.4 ± 3.3 | 83.8 ± 1.0 | 65.7 ± 2.2 | 62.8 ± 3.2 | 77.9 ± 3.3 | 62.3 ± 3.7 | 55.7 ± 3.7 | 78.9 ± 3.1 | 60.6 ± 3.8 | 75.7 ± 5.6 | 69.0 |
| SDE-EDG (OURS) | 70.5 ± 2.9 | 88.8 ± 4.9 | 66.1 ± 2.8 | 55.1 ± 2.6 | 85.1 ± 3.6 | 59.5 ± 4.4 | 58.6 ± 3.3 | 88.2 ± 3.4 | 68.9 ± 3.5 | 72.2 ± 5.7 | 71.3 |

Table 9: PowerSupply. We show the results on each target domain by domain index.

| ALGORITHM | 21 | 22 | 23 | 24 | 25 | 26 | 27 | 28 | 29 | 30 | AVG |
|---|---|---|---|---|---|---|---|---|---|---|---|
| ERM | 69.8 ± 1.4 | 70.0 ± 1.4 | 69.2 ± 1.3 | 64.4 ± 1.5 | 85.8 ± 1.0 | 76.0 ± 1.3 | 70.1 ± 1.5 | 69.8 ± 1.5 | 69.0 ± 1.3 | 65.5 ± 1.5 | 71.0 |
| MIXUP | 69.6 ± 1.4 | 69.5 ± 1.5 | 68.3 ± 1.5 | 64.3 ± 1.5 | 87.1 ± 1.0 | 76.6 ± 1.3 | 70.1 ± 1.4 | 69.2 ± 1.3 | 68.1 ± 1.5 | 65.0 ± 1.6 | 70.8 |
| MMD | 70.0 ± 1.3 | 69.7 ± 1.4 | 68.7 ± 1.4 | 64.8 ± 1.5 | 85.6 ± 1.4 | 76.1 ± 1.3 | 70.0 ± 1.5 | 69.5 ± 1.4 | 68.7 ± 1.3 | 65.6 ± 1.5 | 70.9 |
| MLDG | 69.7 ± 1.4 | 69.7 ± 1.5 | 68.6 ± 1.5 | 64.6 ± 1.5 | 86.4 ± 1.1 | 76.3 ± 1.4 | 70.1 ± 1.4 | 69.4 ± 1.3 | 68.4 ± 1.5 | 65.6 ± 1.5 | 70.8 |
| IRM | 69.8 ± 1.4 | 69.5 ± 1.4 | 68.3 ± 1.4 | 64.1 ± 1.4 | 87.2 ± 0.9 | 76.5 ± 1.3 | 70.0 ± 1.5 | 69.1 ± 1.5 | 68.2 ± 1.3 | 65.0 ± 1.4 | 70.8 |
| RSC | 69.9 ± 1.4 | 69.6 ± 1.4 | 68.6 ± 1.4 | 64.4 ± 1.5 | 86.6 ± 1.0 | 76.3 ± 1.3 | 70.0 ± 1.5 | 69.4 ± 1.4 | 68.4 ± 1.3 | 65.4 ± 1.5 | 70.9 |
| MTL | 69.6 ± 1.4 | 69.4 ± 1.5 | 68.2 ± 1.6 | 64.2 ± 1.5 | 87.4 ± 1.2 | 76.6 ± 1.3 | 69.9 ± 1.5 | 69.1 ± 1.5 | 68.2 ± 1.5 | 64.6 ± 1.4 | 70.7 |
| FISH | 69.7 ± 1.4 | 69.4 ± 1.4 | 68.2 ± 1.4 | 64.2 ± 1.4 | 87.3 ± 1.0 | 76.6 ± 1.3 | 69.9 ± 1.5 | 69.2 ± 1.5 | 68.2 ± 1.3 | 65.2 ± 1.5 | 70.8 |
| CORAL | 69.9 ± 1.4 | 69.7 ± 1.4 | 68.9 ± 1.4 | 64.6 ± 1.4 | 86.1 ± 1.0 | 76.3 ± 1.3 | 70.0 ± 1.5 | 69.5 ± 1.5 | 68.8 ± 1.3 | 65.7 ± 1.5 | 71.0 |
| ANDMASK | 69.9 ± 1.4 | 69.4 ± 1.4 | 68.2 ± 1.3 | 64.0 ± 1.4 | 87.4 ± 0.9 | 76.7 ± 1.3 | 70.0 ± 1.5 | 69.1 ± 1.5 | 68.0 ± 1.3 | 64.7 ± 1.5 | 70.7 |
| DIVA | 69.7 ± 1.4 | 69.5 ± 1.3 | 68.2 ± 1.4 | 63.9 ± 1.5 | 87.5 ± 1.0 | 76.5 ± 1.3 | 69.9 ± 1.5 | 69.1 ± 1.5 | 68.1 ± 1.3 | 64.7 ± 1.5 | 70.7 |
| LSSAE | 70.0 ± 1.4 | 69.8 ± 1.4 | 69.0 ± 1.5 | 65.4 ± 1.4 | 85.1 ± 1.1 | 76.0 ± 1.4 | 70.1 ± 1.7 | 69.9 ± 1.3 | 69.0 ± 1.6 | 66.3 ± 1.4 | 71.1 |
| GI | 70.2 ± 1.4 | 71.0 ± 1.4 | 70.5 ± 1.5 | 69.6 ± 1.5 | 80.7 ± 1.1 | 68.4 ± 1.3 | 72.9 ± 1.5 | 72.0 ± 1.3 | 71.8 ± 1.3 | 66.5 ± 1.5 | 71.4 |
| DDA | 69.8 ± 1.6 | 72.4 ± 1.5 | 70.5 ± 1.5 | 63.8 ± 1.5 | 83.7 ± 1.2 | 70.1 ± 1.3 | 70.1 ± 1.3 | 71.4 ± 1.5 | 70.5 ± 1.7 | 63.4 ± 1.2 | 70.9 |
| DRAIN | 70.1 ± 1.3 | 70.0 ± 1.0 | 69.3 ± 1.1 | 65.5 ± 1.5 | 83.6 ± 1.0 | 75.8 ± 1.7 | 70.3 ± 1.3 | 69.8 ± 1.5 | 68.9 ± 1.9 | 66.4 ± 1.2 | 71.0 |
| SDE-EDG (OURS) | 67.7 ± 1.2 | 74.2 ± 1.5 | 79.3 ± 1.0 | 75.2 ± 1.3 | 87.9 ± 1.0 | 78.0 ± 1.2 | 67.2 ± 1.6 | 72.0 ± 1.5 | 79.8 ± 1.2 | 75.3 ± 1.0 | 75.7 |

# C  EXPERIMENTAL SETUP

## C.1  DATASETS

**Dataset: Rotated Gaussian** (Zeng et al., 2023) consists of 30 domains generated by the same Gaussian distribution, but the decision boundary rotates from $0°$ to $338°$ with an interval of $12°$. We split the domains into source domains (1-22 domains), intermediate domains (22-25 domains), and target domains (26-30 domains). The intermediate domains are utilized as the validation set.

**Dataset: Circle** (Pesaranghader & Viktor, 2016) contains evolving 30 domains where the instance are sampled from 30 2D Gaussian distributions. The label is assigned using a half-circle curve as the decision boundary. (15 source domains, 5 validation domains, and 10 target domains)

Table 10: OcularDisease. We show the results on each target domain by domain index.

| ALGORITHM | 30 | 31 | 32 | 33 | AVG |
|---|---|---|---|---|---|
| ERM | $53.4 \pm 5.8$ | $58.1 \pm 4.8$ | $60.8 \pm 5.6$ | $59.2 \pm 7.4$ | 57.9 |
| MIXUP | $59.1 \pm 4.8$ | $67.8 \pm 2.9$ | $46.7 \pm 4.6$ | $65.4 \pm 5.8$ | 59.7 |
| MMD | $58.0 \pm 5.2$ | $56.1 \pm 7.0$ | $66.7 \pm 7.2$ | $49.6 \pm 6.5$ | 57.6 |
| MLDG | $46.6 \pm 6.0$ | $57.8 \pm 6.6$ | $30.0 \pm 5.7$ | $41.2 \pm 7.4$ | 43.9 |
| IRM | $39.8 \pm 7.2$ | $41.4 \pm 4.3$ | $53.3 \pm 6.8$ | $50.4 \pm 4.0$ | 46.2 |
| RSC | $43.2 \pm 6.7$ | $59.4 \pm 4.7$ | $49.2 \pm 6.5$ | $66.2 \pm 7.7$ | 54.5 |
| MTL | $58.0 \pm 6.5$ | $61.9 \pm 6.3$ | $56.7 \pm 14.1$ | $62.1 \pm 9.3$ | 59.7 |
| FISH | $50.0 \pm 6.2$ | $43.6 \pm 6.6$ | $39.2 \pm 4.3$ | $60.0 \pm 8.0$ | 48.2 |
| CORAL | $52.3 \pm 7.0$ | $68.6 \pm 4.6$ | $54.2 \pm 5.1$ | $65.4 \pm 6.5$ | 60.1 |
| ANDMASK | $42.0 \pm 7.1$ | $65.6 \pm 6.9$ | $50.0 \pm 7.1$ | $47.1 \pm 6.7$ | 51.2 |
| DIVA | $51.1 \pm 7.1$ | $53.1 \pm 4.2$ | $51.7 \pm 12.2$ | $56.7 \pm 4.6$ | 53.1 |
| LSSAE | $55.0 \pm 4.8$ | $45.3 \pm 3.8$ | $57.5 \pm 8.2$ | $51.2 \pm 5.9$ | 52.3 |
| DDA | $53.4 \pm 7.9$ | $63.6 \pm 3.1$ | $44.2 \pm 6.0$ | $62.1 \pm 5.1$ | 55.8 |
| DRAIN | $58.0 \pm 6.9$ | $62.8 \pm 6.4$ | $53.3 \pm 6.8$ | $60.8 \pm 9.8$ | 58.7 |
| SDE-EDG (OURS) | $54.5 \pm 6.7$ | $66.4 \pm 5.9$ | $64.2 \pm 8.8$ | $65.4 \pm 8.7$ | 62.6 |

Table 11: Neural network architectures for different datasets. (MLP is short for Multiple-layer Perceptrons)

| DATASET | FEATURE EXTRACTOR | CLASSIFIER | $f$ & $g$ |
|---|---|---|---|
| RGAUSSIAN | $[2, 32, 32]$-MLP | A LINEAR LAYER | $[32, 32]$-MLP |
| CIRCLE | $[2, 32, 32]$-MLP | A LINEAR LAYER | $[32, 32]$-MLP |
| SINE | $[2, 32, 32]$-MLP | A LINEAR LAYER | $[32, 32]$-MLP |
| ROTATING MNIST | MNIST CONVNET | A LINEAR LAYER | $[128, 512, 512]$-MLP |
| PORTRAIT | RESNET-18 | A LINEAR LAYER | $[512, 512]$-MLP |
| CALTRAN | RESNET-18 | A LINEAR LAYER | $[512, 512]$-MLP |
| POWERSUPPLY | $[2, 256, 256]$-MLP | A LINEAR LAYER | $[256, 256]$-MLP |
| OCULARDISEASE | RESNET-18 | A LINEAR LAYER | $[512, 512]$-MLP |

**Dataset: Sine** In Sine (Pesaranghader & Viktor, 2016) each data owns two attributes $(x_1, x_2)$. The label is assigned using a sine curve as the decision boundary. We rearrange this dataset by extending it to 24 evolving domains. Each domain covers $\frac{1}{24}$ the period of the sinusoid. (12 source domains, 4 validation domains, and 8 target domains)

**Dataset: Rotated MNIST (RMNIST)** (Ghifary et al., 2015) is an adaptation of the popular MNIST digit dataset (Deng, 2012), composed of MNIST digits of various rotations. The task is to classify a digit from 0 to 9 given an image of the digit. We follow (Qin et al., 2022) and extend it to 19 evolving domains via applying the rotations with degree of $\{0°, 15°, 30°, \dots, 180°\}$ in order. (10 source domains, 3 validation domains, and 6 target domains).

**Dataset: Portraits** (Ginosar et al., 2015) is a real-world dataset that comprises photographs of American high school seniors collected over a period of 108 years (1905-2013) across 26 states. The objective is to accurately classify the gender for each photograph. The dataset is divided into 34 domains based on a fixed interval over time. (19 source domains, 5 validation domains, and 10 target domains)

**Dataset: Caltran** (Hoffman et al., 2014) consists of real-world images captured by a fixed traffic camera deployed in an intersection over time. Frames were updated at 3-minute intervals each with a resolution $320 \times 320$. We divide it into 34 domains by time. The task of Caltran is to classify scenes to identify the presence of one or more vehicles in or approaching the intersection. The challenge mainly raise from the continually evolving domain shift as changes include time, illumination, weather, etc. (19 source domains, 5 validation domains, and 10 target domains)

**Dataset: PowerSupply** (Dau et al., 2019) is a dataset designed for the task of time-section prediction of current power supply based on hourly records obtained from an Italian electricity company. The dataset consists of 30 domains formed according to days. Each data point is assigned a binary class label indicating whether the current power supply belongs to the morning or the afternoon. Domain

Table 12: Hyper-parameters and selected values

| DATASET | PARAMETERS | VALUE |
|---|---|---|
| RGAUSSIAN | $\alpha$ | 10 |
| | LEARNING RATE | 1e-3 |
| | BATCH SIZE | 64 |
| CIRCLE | $\alpha$ | 10 |
| | LEARNING RATE | 1e-3 |
| | BATCH SIZE | 64 |
| SINE | $\alpha$ | 10 |
| | LEARNING RATE | 1e-3 |
| | BATCH SIZE | 64 |
| RMNIST | $\alpha$ | 1 |
| | LEARNING RATE | 1e-3 |
| | BATCH SIZE | 48 |
| PORTRAITS | $\alpha$ | 1 |
| | LEARNING RATE | 1e-4 |
| | BATCH SIZE | 24 |
| CALTRAN | $\alpha$ | 1 |
| | LEARNING RATE | 5e-5 |
| | BATCH SIZE | 24 |
| POWERSUPPLY | $\alpha$ | 10 |
| | LEARNING RATE | 1e-3 |
| | BATCH SIZE | 64 |
| OCULARDISEASE | $\alpha$ | 0.5 |
| | LEARNING RATE | 5e-5 |
| | BATCH SIZE | 24 |

shifts may arise due to variations in season, weather, price, or the differences between working days and weekends. (15 source domains, 5 validation domains, and 10 target domains)

**Dataset: OcularDisease** (Kaggle, 2020) (from the Kaggle Competition (Kaggle 2020)) Ocular Disease Intelligent Recognition (ODIR) is a structured ophthalmic database of 5,000 patients with age, color fundus photographs from left and right eyes and doctors' diagnostic keywords from doctors. We set three classes: Normal, Diabetes and other diseases. To generate non-stationary environments, we sort the photographs in ascending order of the age of the patients. (27 source domains, 2 validation domains, and 4 target domains)

### C.2 EXPERIMENT SETTING AND IMPLEMENTATION DETAILS

Neural network architectures used for different datasets in Table 11. ResNet18, and MNIST ConvNet are from domainbed codes (Gulrajani & Lopez-Paz, 2020). Specifically, Table 11 demonstrates the classifier architecture for baselines equipped with learned classifier. SDE-EDG does not equip with a classifier network, since the prediction is based on the closest distance to the Evolving Centroids.

We list the values of the hyper-parameters of SDE-EDG for different datasets in Table 12. We use torchsde (Kidger et al., 2021) package for SDE implementations and the step size is 0.05.

For SDE-EDG, to address the issue of lacking access to data at $t_0$, we adopt a sampling strategy that $z_0^k$ is sampled from $\mathcal{N}(\mu_0^k, \sigma_0 I)$ using the Reparameterization trick, where $\mu_0^k$ is a learned parameter and $\sigma_0$ is a predefined constant.

## D THEORETICAL ANALYSIS

In this section, we show $\mathcal{J}_{mle}$ can be decomposed into sums of Path Alignment Loss for stochastic path fittings, and Stochastic Uncertainty Loss for minimizing diffusion terms at observations. Furthermore, we prove that SDE-EDG learning methodology leads to a lower generalization bound.

**Lemma D.1.** *The optimization loss $\mathcal{J}_{mle}$ defined in Eq. (7), is of the upper bound of the Path Alignment loss $\mathcal{J}_{pa}$ and the Stochastic Uncertainty loss $\mathcal{J}_{su}$, which implies (the proof in Appendix E.5)*

$$\mathcal{J}_{mle} \geq \underbrace{\sum_{m=1}^{M}\sum_{k=1}^{K}\sum_{i=1}^{N_B} \frac{1}{2MKN_B} \frac{(z_{m+1} - \hat{z}_{m+1})^2}{g(z_m)^2 \Delta t_m}}_{\mathcal{J}_{pa}} + \underbrace{\sum_{m=1}^{M}\sum_{k=1}^{K}\sum_{i=1}^{N_B} \frac{1}{2MKN_B} \log(2\pi g(z_m)^2 \Delta t_m)}_{\mathcal{J}_{su}}$$

*where we hide the subscript, sample index i, for $z_{m|i}$ as $z_m$ for simplicity.*

Lemma D.1 shows (1) The term $\mathcal{J}_{pa}$ is included in Eq. (13). Thus, minimizing $\mathcal{J}_{pa}$ can lead to a smaller $KL(\nu||\hat{\nu})$ and eventually help to minimize the expected generalization error in Eq. (12). (2) The Stochastic Uncertainty loss $\mathcal{J}_{su}$ serves as a loss function to optimize the diffusion terms $g$. As domains at timestamps $\{t_1, \ldots, t_m, \ldots, t_M\}$ are observed, the stochastic uncertainty depicted by diffusion term at these timestamps should be small (Kong et al., 2020).

**Remark D.2.** (adapted from (Shui et al., 2022)) Let $\nu$ be the stochastic path of unseen target evolving domains sampled sequentially from an evolving environment $\mathcal{E}$ between timestamps from $T$ to $T + T^*$, and $\hat{\nu}$ be the learned stochastic path by our SDE-EDG. Then, we have

$$R_\nu(h) \leq R_{\hat{\nu}}(h) + \frac{G}{\sqrt{2}}\sqrt{\mathrm{KL}(\nu||\hat{\nu})} \tag{12}$$

The proof is in E.2. To achieve a low risk on $\nu$, Remark D.2 suggests learning SDE-EDG to minimize the KL divergence between $\hat{\nu}$ and $\nu$, while $R_{\hat{\nu}}(h)$ can be approximated by the empirical risk.

In this analysis, we assume that $D(t)$ follows $\mathcal{U}(0, T + T^*)$ and $\mathcal{D}(y)$ is invariant through time In this theorem, we analyze the simple case for constant time intervals $\Delta t$ between consecutive domains.

**Theorem D.3.** *For any $\epsilon > 0$, with probability at least $1 - \epsilon$, the KL-divergence on path space of $\nu$ and $\hat{\nu}$ can be upper bounded by:*

$$KL(\nu||\hat{\nu}) \leq \frac{L}{2}\left(\mathcal{J}_{pa} + 2\mu\hat{\mathfrak{R}}_{\mathcal{H}} + 3M\sqrt{\frac{\Delta t \log 2/\epsilon}{2TN}}\right) \tag{13}$$

*where $\hat{\mathfrak{R}}_{\mathcal{H}}$ is the Rademacher complexity of the hypothesis set $\mathcal{H}$. $M > 0$ is the upper bound of $\ell = \frac{(z_{m+1} - \hat{z}_{m+1})^2}{g(z_m)^2 \Delta t_m}$ and $\ell$ is $\mu$-Lipschitz for some $\mu > 0$ for any fixed $z_m$ and $z_{m+1}$ ($\hat{z}_{m+1}$ is mapped from $z_m$).*

The proof is in Appendix E.6. Theorem D.3 suggests: (1) $L$ corresponds to the extrapolation steps on future time. The bound is minimized when $L \to 0$. LHS gets larger with a larger $L$, which indicates the EDG generalizes worse with more future steps. (2) The third term on the RHS gets smaller with a smaller $\Delta t$. It implies we should collect sufficient and diverse source domains to ensure that temporal dynamics can be learned in the learning process. It also indicates that generating continuous-interpolated samples will help the learning process. If we collect only one domain even with an infinite number of samples, obviously EDG can not be achieved; Equally, we can collect a longer time horizon in the source domains, which means a larger $T$ will contribute to better learning evolving patterns and thus a smaller generalization error. A bigger sample size $N$ in each source domain also contributes to a smaller generalization error.

## E    PROOF OF THEORIES

We first prove an intermediate lemma:

**Lemma E.1.** *(Shui et al., 2022) Let $v \in \mathcal{V} = \mathcal{X} \times \mathcal{Y}$ be the real-valued integrable random variable, let $P$ and $Q$ be two distributions on a common space $\mathcal{V}$ such that $Q$ is absolutely continuous w.r.t. $P$. If for any function $f$ and $\lambda \in \mathbb{R}$ such that $\mathbb{E}_P[e^{\lambda(f(v) - \mathbb{E}_P(f(v))}] < \infty$, then we have:*

$$\lambda(\mathbb{E}_Q f(v) - \mathbb{E}_P f(v)) \leq D_{KL}(Q||P) + \log \mathbb{E}_P[e^{\lambda(f(v) - \mathbb{E}_P(f(v)))}],$$

*where $D_{KL}(Q||P)$ is the Kullback–Leibler divergence between distribution $Q$ and $P$, and the equality arrives when $f(v) = \mathbb{E}_P f(v) + \frac{1}{\lambda}\log(\frac{dQ}{dP})$.*

*Proof.* We let $g$ be any function such that $\mathbb{E}_P[e^{g(v)}] < \infty$, then we define a random variable $V_g(v) = \frac{e^{g(v)}}{\mathbb{E}_P[e^{g(v)}]}$, then we can verify that $\mathbb{E}_P(V_g) = 1$. We assume another distribution $Q$ such that

$Q$ (with distribution density $q(v)$) is absolutely continuous w.r.t. $P$ (with distribution density $p(v)$), then we have:

$$\mathbb{E}_Q[\log V_g] = \mathbb{E}_Q[\log \frac{q(v)}{p(v)} + \log(V_g \frac{p(v)}{q(v)})] = D_{\text{KL}}(Q\|P) + \mathbb{E}_Q[\log(V_g \frac{p(v)}{q(v)})]$$

$$\leq D_{\text{KL}}(Q\|P) + \log \mathbb{E}_Q[\frac{p(v)}{q(v)} V_g] = D_{\text{KL}}(Q\|P) + \log \mathbb{E}_P[V_g]$$

Since $\mathbb{E}_P[V_g] = 1$ and according to the definition we have $\mathbb{E}_Q[\log V_g] = \mathbb{E}_Q[g(v)] - \mathbb{E}_Q \log \mathbb{E}_P[e^{g(v)}] = \mathbb{E}_Q[g(v)] - \log \mathbb{E}_P[e^{g(v)}]$ (since $\mathbb{E}_P[e^{g(v)}]$ is a constant w.r.t. $Q$) and we therefore have:

$$\mathbb{E}_Q[g(v)] \leq \log \mathbb{E}_P[e^{g(v)}] + D_{\text{KL}}(Q\|P) \tag{14}$$

Since this inequality holds for any function $g$ with finite moment generation function, then we let $g(v) = \lambda(f(v) - \mathbb{E}_P f(v))$ such that $\mathbb{E}_P[e^{f(v) - \mathbb{E}_P f(v)}] < \infty$. Therefore we have $\forall \lambda$ and $f$ we have:

$$\mathbb{E}_Q \lambda(f(v) - \mathbb{E}_P f(v)) \leq D_{\text{KL}}(Q\|P) + \log \mathbb{E}_P[e^{\lambda(f(v) - \mathbb{E}_P f(v))}]$$

Since we have $\mathbb{E}_Q \lambda(f(v) - \mathbb{E}_P f(v)) = \lambda \mathbb{E}_Q(f(v) - \mathbb{E}_P f(v))) = \lambda(\mathbb{E}_Q f(v) - \mathbb{E}_P f(v))$, therefore we have:

$$\lambda(\mathbb{E}_Q f(v) - \mathbb{E}_P f(v)) \leq D_{\text{KL}}(Q\|P) + \log \mathbb{E}_P[e^{\lambda(\mathbb{E}_Q f(v) - \mathbb{E}_P f(v))}]$$

As for the attainment in the equality of Eq. (14), we can simply set $g(v) = \log(\frac{q(v)}{p(v)})$, then we can compute $\mathbb{E}_P[e^{g(v)}] = 1$ and the equality arrives. Therefore in Lemma 1, the equality reaches when $\lambda(f(v) - \mathbb{E}_P f(v)) = \log(\frac{dQ}{dP})$. $\qquad\square$

In the classification problem, we define the observation pair $v = (x, y)$. We also define the loss function $\ell(v) = L \circ h(v)$ with deterministic hypothesis $h$ and prediction loss function $L$. Then for abuse of notation, we simply denote the loss function $\ell(v)$ in this part.

Then we introduce the following bound between SDE-EDG's stochastic path $\hat{\nu}$ and real evolving path $\nu$.

**Remark E.2. (Restatement of Remark D.2)** Let $\hat{\nu}$ be the learned stochastic path by SDE-EDG, and suppose the loss function $\ell$ is bounded within an interval $G : G = \max(\ell) - \min(\ell)$. Then, for any $h \in \mathcal{H}$, its target risk $R_\nu(h)$ can be upper bounded by:

$$R_\nu(h) \leq R_{\hat{\nu}}(h) + \frac{G}{\sqrt{2}}\sqrt{\text{KL}(\nu\|\hat{\nu})}.$$

where we use $\text{KL}(\cdot\|\cdot)$ to denote the KL divergence for simplification in the remaining paragraphs.

*Proof.* According to Lemma E.1, $\forall \lambda > 0$ we have:

$$\mathbb{E}_Q f(v) - \mathbb{E}_P f(v) \leq \frac{1}{\lambda}(\log \mathbb{E}_P e^{[\lambda(f(v) - \mathbb{E}_P f(v))]} + D_{\text{KL}}(Q\|P)) \tag{15}$$

And $\forall \lambda < 0$ we have:

$$\mathbb{E}_Q f(v) - \mathbb{E}_P f(v) \geq \frac{1}{\lambda}(\log \mathbb{E}_P e^{[\lambda(f(v) - \mathbb{E}_P f(v))]} + D_{\text{KL}}(Q\|P)) \tag{16}$$

Let $f = \ell$. Since the random variable $\ell$ is bounded through $G = \max(\ell) - \min(\ell)$, then according to (Wainwright, 2019) (Chapter 2.1.2), $\ell - \mathbb{E}_P \ell$ is sub-Gaussian with parameter at most $\sigma = \frac{G}{2}$, then we can apply Sub-Gaussian property to bound the $\log$ moment generation function:

$$\log \mathbb{E}_P e^{[\lambda(\ell(v) - \mathbb{E}_P \ell(v))]} \leq \log e^{\frac{\lambda^2 \sigma^2}{2}} \leq \frac{\lambda^2 G^2}{8}.$$

In Eq. (15), we let $Q = \mathcal{D}'$ and $P = \mathcal{D}$, then $\forall \lambda > 0$ we have:

$$\mathbb{E}_{\mathcal{D}'}\ell(v) - \mathbb{E}_\mathcal{D}\ell(v) \leq \frac{1}{\lambda}D_{\text{KL}}(\mathcal{D}'\|\mathcal{D}) + \frac{G^2\lambda}{8} \tag{17}$$

Since the inequality holds for $\forall \lambda$, then by taking $\lambda = \frac{2\sqrt{2}}{G}\sqrt{D_{\mathrm{KL}}(\mathcal{D}'\|\mathcal{D})}$ we finally have:

$$\mathbb{E}_{\mathcal{D}'}\,\ell(v) \leq \mathbb{E}_{\mathcal{D}}\,\ell(v) + \frac{G}{\sqrt{2}}\sqrt{D_{\mathrm{KL}}(\mathcal{D}'\|\mathcal{D})} \tag{18}$$

Let $\mathcal{D}' = \nu$ and $\mathcal{D} = \hat{\nu}$, we complete our proof.

$\square$

**Lemma E.3.** *Define* $\mathcal{R}_{pa} = \mathbb{E}\frac{\|z_{t+\Delta t} - \hat{z}_{t+\Delta t}\|^2}{2g_k(z_t)^2\Delta t}$ *as the expected of the model parameterized by $\theta$ with the sample $\{z, y, t\}$, which is upper bounded by $M > 0$ and is $\mu$-Lipschitz for some $\mu > 0$ for any fixed $z_t^k$ and $\hat{z}_t^k$. Then, for any $\epsilon > 0$, with probability at least $1 - \epsilon$, the following inequalities holds*

$$\mathcal{R}_{pa} \leq \mathcal{J}_{pa} + 2\mu\hat{\mathfrak{R}}_{\mathcal{H}} + 3M\sqrt{\frac{\Delta t \log 2/\epsilon}{2TN}} \tag{19}$$

*where $\hat{\mathfrak{R}}_{\mathcal{H}}$ is the Rademacher complexity of the hypothesis set $\mathcal{H}$.*

Lemma E.3 is directly adapted from Theorem 11.3 in (Mohri et al., 2018).

**Assumption E.4.** (Realizable) Assume there exists a hypothesis $h^* \in \mathcal{H}$ that satisfies

$$z_{m+1}^k = z_m^k + \int_{t_m}^{t_{m+1}} \zeta_k(z_s^k, s)ds + \int_{t_m}^{t_{m+1}} g_k(z_s^k, s)dB_s, \tag{20}$$

where $\zeta_k$ is the drift function of $h^*$ and $g_k$ is the diffusion function of $h^*$. Given the same diffusion function shared in Eq. (6) and Eq. (20), the KL between them on path space can be approximated (Xu et al., 2022; Li et al., 2020).

**Lemma E.5.** *(Restatement of Lemma D.1) The optimization loss $\mathcal{J}_{mle}$ defined in Eq. (7), is of the upper bound of the Path Alignment loss $\mathcal{J}_{pa}$ and the Stochasti Uncertainty loss $\mathcal{J}_{su}$, which implies*

$$\mathcal{J}_{mle} \geq \underbrace{\sum_{m=1}^{M}\sum_{k=1}^{K}\sum_{i=1}^{N_B} \frac{1}{2MKN_B}\frac{(z_{m+1}^k - \hat{z}_{m+1}^k)^2}{g_k(z_m)^2\Delta t_m}}_{\mathcal{J}_{pa}} + \underbrace{\sum_{m=1}^{M}\sum_{k=1}^{K}\sum_{i=1}^{N_B}\frac{1}{2MKN_B}\log(2\pi g_k(z_m^k)^2\Delta t_m)}_{\mathcal{J}_{sc}}$$

*Proof.* We did not write the superscript $k$ in the proof for simplicity. Let $z_m, \hat{z}_{m+1}$ generated by the Euler discretization:

$$\hat{z}_{m+1} = z_m + f(z_m)\Delta t_m + g(z_m)(B_{m+1} - B_m) = z_m + f(z_m)\Delta t_m + g(z_m)\Delta t_m^{1/2}\epsilon_{m+1} \tag{21}$$

where $\{B_m\}_{t_m \geq 0}$ is the Brownian motion, and $\epsilon_m$ is sampled from $\mathcal{N}(0, I)$. This implies that conditional on the previous state, the current state is normally distributed: $\hat{\mathcal{D}}(z_{m+1}|z_m) = \mathcal{N}(z_m + f(z_m)\Delta t_m, g(z_m)^2\Delta t_m)$. Thus, the log-densities can be evaluated as

$$\begin{aligned} MKN_B \cdot \mathcal{J}_{mle} &= \sum_{m=1}^{M}\sum_{k=1}^{K}\sum_{i=1}^{N_B} -\log\hat{\mathcal{D}}\big(z = \hat{z}_{m+1|i}^k \big| z = z_{m|i}^k\big) = \sum_{m=1}^{M}\sum_{k=1}^{K}\sum_{i=1}^{N_B} -\mathbb{E}\log\hat{\mathcal{D}}(z_{m+1}|z_m) \\ &= \sum_{m=1}^{M}\sum_{k=1}^{K}\sum_{i=1}^{N_B}\left[\frac{1}{2}\log(2\pi g(z_m)^2\Delta t_m) + \frac{1}{2}\frac{(z_{m+1} - (z_m + f(z_m)\Delta t_m))^2}{g(z_m)^2\Delta t_m}\right] \\ &= \sum_{m=1}^{M}\sum_{k=1}^{K}\sum_{i=1}^{N_B}\left[\frac{1}{2}\log(2\pi g(z_m)^2\Delta t_m) + \frac{1}{2}\frac{(z_{m+1} - \hat{z}_{m+1} + g(z_m)\Delta t_m^{1/2}\epsilon_{m+1})^2}{g(z_m)^2\Delta t_m}\right] \\ &= \sum_{m=1}^{M}\sum_{k=1}^{K}\sum_{i=1}^{N_B}\left[\frac{1}{2}\log(2\pi g(z_m)^2\Delta t_m) + \frac{1}{2}\frac{(z_{m+1} - \hat{z}_{m+1})^2}{g(z_m)^2\Delta t_m} + \frac{1}{2}\epsilon_{m+1}^2\right. \\ &\quad \left. + \frac{(z_{m+1} - \hat{z}_{m+1})\epsilon_{m+1}}{g(z_m)\Delta t_m^{1/2}}\right] \end{aligned}$$

where the second equality is because $\tilde{z}_{m+1|i}^k$ is the feature of a real sample and we hide the subscript index $i$ for simplicity.

$$\sum_{m=1}^{M} \sum_{k=1}^{K} \sum_{i=1}^{N_B} \frac{1}{2} \epsilon_{m+1}^2 \approx \frac{1}{2} \mathbb{E} \epsilon_{m+1}^2 = 1/2$$

where we applies that $\epsilon_m$ follows normal distribution.

$$\sum_{m=1}^{M} \sum_{k=1}^{K} \sum_{i=1}^{N_B} \frac{(z_{m+1} - \hat{z}_{m+1}) \epsilon_{m+1}}{g(z_m) \Delta t_m^{1/2}} \approx \mathbb{E} \frac{(f(z_m) - \zeta(z_m)) \Delta t_m^{1/2} \epsilon_{m+1}}{g(z_m)} = 0 \tag{22}$$

where we apply the above equation is martingale and its expectation is zero (Xu et al., 2022; Li et al., 2020). Then, we conclude our proof.

$\square$

**Theorem E.6.** *The KL-divergence on path space of $\nu$ and $\hat{\nu}$ is upper bounded by:*

$$KL(\nu||\hat{\nu}) \leq \frac{L}{2} \left( \mathcal{J}_{pa} + 2\mu \hat{\mathfrak{R}}_{\mathcal{H}} + 3M \sqrt{\frac{\Delta t \log 2/\epsilon}{2TN}} \right) \tag{23}$$

*Proof.*

$$\begin{aligned}
\text{KL}(\nu||\hat{\nu}) &= \mathbb{E}_\nu \big[ \log \prod_{l=1}^{L} \mathcal{D}(z_{l+1}, y_{l+1}|z_l) - \log \prod_{l=1}^{L} \hat{\mathcal{D}}(z_{l+1}, y_{l+1}|z_l) \big] \\
&= \mathbb{E}_\nu \big[ \log \prod_{l=1}^{L} \frac{\mathcal{D}(z_{l+1}|z_l, y_{l+1})}{\hat{\mathcal{D}}(z_{l+1}|z_l, y_{l+1})} + \log \prod_{l=1}^{L} \frac{\mathcal{D}(y_{l+1})}{\hat{\mathcal{D}}(y_{l+1})} \big] \\
&= \mathbb{E}_\nu \log \prod_{l=1}^{L} \frac{\mathcal{D}(z_{l+1}|z_l, y_{l+1})}{\hat{\mathcal{D}}(z_{l+1}|z_l, y_{l+1})}
\end{aligned}$$

where in the first line, we replace the KL-divergence corresponding to the diagram of data generation process and $\mathcal{D}(t)$ following $\mathcal{U}(0, T + T^*)$ cancels out the KL-divergence of $t$ between $\mathcal{D}$ and $\hat{\mathcal{D}}$; in the second equation, we use the independence of $y_{M+l+1}$ to $z_{M+l}$ by d-separation (Bishop & Nasrabadi, 2006) from figure 1.

Let $z_{M+1}, z_{M+2}, \ldots, z_{M+L}$ generated by the Euler discretization:

$$\begin{aligned}
z_{M+l+1} &= z_{M+l} + \zeta(z_{M+l}) \Delta t_l + g(z_{M+l})(B_{M+l+1} - B_{M+l}) \\
&= z_{M+l} + \zeta(z_{M+l}) \Delta t_l + g(z_{M+l}) \Delta t_l^{1/2} \epsilon_{M+l+1} \tag{24}
\end{aligned}$$

where $\{B_t\}_{t \geq 0}$ is the Brownian motion, and $\epsilon_t$ is sampled from $\mathcal{N}(0, I)$. This implies that conditional on the previous state, the current state is normally distributed: $z_{M+l+1}|z_{M+l} \sim \mathcal{N}(z_{M+l} + \zeta(z_{T+l}) \Delta t_l, g(z_{M+l})^2 \Delta t_l)$. Thus, the log-densities can be evaluated as

$$\log \mathcal{D}(z_{M+l+1}|z_{M+l}) = -\frac{1}{2} \log(2\pi g(z_{M+l})^2 \Delta t_l) - \frac{1}{2} \frac{(z_{M+l} - (z_{M+l} + \zeta(z_{M+l}) \Delta t_l))^2}{g(z_{M+l})^2 \Delta t_l} \tag{25}$$

where $i = 0, 1, \cdots, L$. On the other hand, if at any time step, the state in the next domain was generated by observing $\mathcal{D}_t$, we would have the following log-densities:

$$\log \hat{\mathcal{D}}(z_{M+l+1}|z_{M+l}) = -\frac{1}{2} \log(2\pi g(z_{M+l})^2 \Delta t_l) - \frac{1}{2} \frac{(z_{M+l+1} - (z_{M+l} + f(z_{M+l}) \Delta t_l))^2}{g(z_{M+l})^2 \Delta t_l} \tag{26}$$

Now, we replace $z_{M+l+1}$ in Eq. (24) into Eq. (25) and Eq. (26),

$$\begin{aligned}
\log \mathcal{D}(z_{M+l+1}|z_{M+l}) &= -\frac{1}{2} \log(2\pi g(z_{M+l})^2 \Delta t_l) - \frac{1}{2} \epsilon_{M+l+1}^2 \\
\log \hat{\mathcal{D}}(z_{M+l+1}|z_{M+l}) &= -\frac{1}{2} \log(2\pi g(z_{M+l})^2 \Delta t_l) - \frac{1}{2} \bigg( \frac{(\zeta(z_{M+l}) - f(z_{M+l}))^2}{g(z_{M+l})^2} \Delta t_l \\
&\quad + \frac{2(\zeta(z_{M+l}) - f(z_{M+l})) \epsilon_{M+l+1}}{g(z_{M+l})} \Delta t_l^{1/2} + \epsilon_{M+l+1}^2 \bigg)
\end{aligned}$$

The KL divergence on the path space could then be regarded as a sum of infinitely many KL-divergences between Gaussians (applying $\Delta t_l = \Delta t$):

$\text{KL}(\nu||\hat{\nu})$

$=\mathbb{E}_\nu \lim_{\Delta t \to 0} \sum_{l=0}^{T^*/\Delta t} \mathbb{E}_{z_{M+l}} \left[ \text{KL}(\mathcal{D}(z_{M+l+1}|z_{M+l}, y=k)||\hat{\mathcal{D}}(z_{M+l+1}|z_{M+l}, y=k)) \right]$

$=\mathbb{E}_\nu \lim_{\Delta t \to 0} \sum_{l=0}^{T^*/\Delta t} \mathbb{E}_{z_{M+l}^k} \mathbb{E}_{z_{M+l+1}^k} \log \frac{\mathcal{D}(z_{M+l+1}|z_{M+l}, y=k)}{\hat{\mathcal{D}}(z_{M+l+1}|z_{M+l}, y=k)}$

$=\mathbb{E}_\nu \lim_{\Delta t \to 0} \sum_{l=0}^{T^*/\Delta t} \mathbb{E}_{z_{M+l}^k} \mathbb{E}_{\epsilon_{M+l+1}} \frac{(\zeta_k(z_{M+l}^k) - f_k(z_{M+l}^k))^2}{2g_k(z_{M+l})^2}\Delta t_l + \frac{(\zeta_k(z_{M+l}^k) - f_k(z_{M+l}^k))}{g_k(z_{M+l})}\Delta t_l^{1/2}\epsilon_{M+l+1}$

$=\mathbb{E}_\nu \lim_{\Delta t \to 0} \sum_{l=0}^{T^*/\Delta t} \frac{\Delta t}{2}\mathbb{E}_{z_t^k}(u_t^k)^2 + \int_T^{T+T^*} u_t^k \mathrm{d}B_t$

$=\mathbb{E}_\nu \lim_{\Delta t \to 0} \sum_{l=0}^{T^*/\Delta t} \frac{\Delta t}{2}\mathbb{E}_{z_t^k}(u_t^k)^2$

where $u_t^k = (\zeta_k(z_t^k, t) - f_k(z_t^k, t))/g_k(z_t)$, and $t \in \{t_1, \ldots, t_{M+L}\}$. In the last equality, we used the fact that the Itô integral $\int_T^{T+T^*} u_t^k \mathrm{d}B_t$ is a martingale.

$$\mathbb{E}_\nu \lim_{\Delta t \to 0} \sum_{l=0}^{T^*/\Delta t} \frac{\Delta t}{2}\mathbb{E}_{z_t^k}\left(\frac{\zeta_k(z_t^k) - f_k(z_t^k)}{g_k(z_t)}\right)^2 = \frac{T^*}{2}\mathbb{E}_\mathcal{D}\left(\frac{\zeta_k(z_t^k) - f_k(z_t^k)}{g_k(z_t)}\right)^2$$

(27)

With fixed interval $t_{m+1} - t_m = t_{M+l+1} - t_{M+l} = \Delta t$, we have

$$z_{t+\Delta t}^k = z_t^k + \zeta_k(z_t^k, t)\Delta t + g_k(z_t^k, t_t)(B_{t+\Delta t} - B_t)$$
$$\hat{z}_{t+\Delta t}^k = z_t^k + f_k(z_t^k, t)\Delta t + g_k(z_t^k, t)(B_{t+\Delta t} - B_t)$$

Hence we have

$\text{KL}(\nu||\hat{\nu})$

$\leq T^* \mathbb{E} \frac{1}{2g_k(z_t)^2 \Delta t^2}\left( \left|\left|[z_t^k + \zeta_k(z_t^k)\Delta t + g_k(z_t^k, t)(B_{t+\Delta t} - B_t)] - [z_t^k + f_k(z_t)\Delta t + g_k(z_t^k, t)(B_{t+\Delta t} - B_t)]\right|\right|^2 \right)$

$= L \cdot \mathbb{E} \frac{||z_{t+\Delta t} - \hat{z}_{t+\Delta t}||^2}{2g_k(z_t)^2 \Delta t}$

Hence, we conclude the proof.

$\square$

## F  ADDITIONAL EXPERIMENT RESULTS

**Evolving Domain Generalization with Continuous Index**  We evaluate the performance of our method on the continuous index cases by modifying the rotation degrees of the target domains in the RMNIST dataset as $\{129.1°, 137.6°, 144.3°, 149.2°, 155.5°\}$. We conduct experiments and compare our method with GI, which can also work with continuous index. Compared to GI incremental improvements ($0.7\%$) on RMNIST, SDE-EDG could achieve a significant improvement with $12.9\%$ higher accuracy on ERM. This shows the superiority of our proposed approach not only in handling discrete index EDG tasks, but also the continuous cases.

**Rotated Gaussian Results**  The prediction results of the last 4 target domains (27-30) are visualized in Figure 5. Rotated Gaussian has a fixed margin $p(x)$ but an evolving $p(y|x)$. ERM has been biased by the source domains, and it fails for all target domains by making the opposite labelings. Instead, SDE-EDG could give the correct predictions capturing the temporal patterns (rotating decision boundary by $12°$ with time).

Table 13: Rotated MNIST (RMNIST) with the Random Continuous Index

| ALGORITHM | 129.1° | 137.6° | 144.3° | 149.2° | 155.5° | AVG |
|---|---|---|---|---|---|---|
| ERM | 57.8 ± 0.9 | 45.0 ± 0.9 | 40.6 ± 0.8 | 40.0 ± 0.9 | 38.3 ± 0.8 | 44.3 |
| GI | 59.0 ± 0.8 | 45.7 ± 0.7 | 41.8 ± 0.8 | 39.6 ± 0.8 | 38.9 ± 0.7 | 45.0 |
| SDE-EDG (OURS) | **76.2 ± 0.7** | **62.9 ± 0.8** | **54.3 ± 0.8** | **49.6 ± 0.8** | **42.8 ± 0.8** | **57.2** |

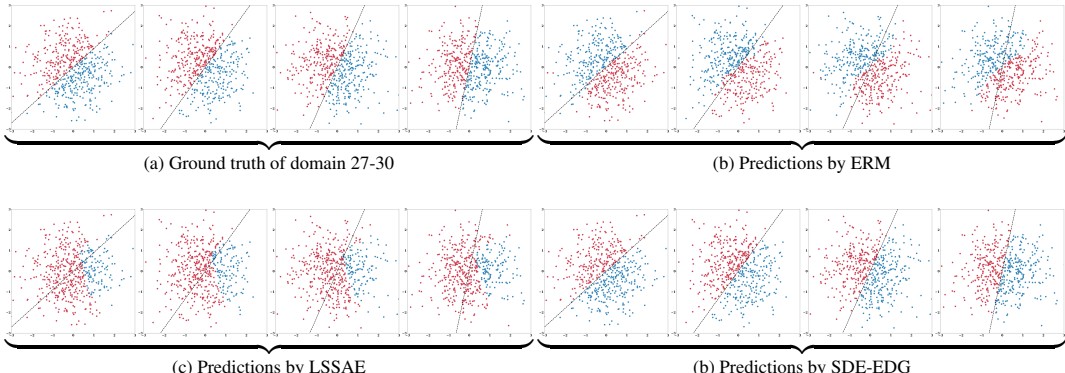

(a) Ground truth of domain 27-30

(b) Predictions by ERM

(c) Predictions by LSSAE

(b) Predictions by SDE-EDG

Figure 5: (a) The ground truth of the Rotated Gaussian between domains 27-30, and positive and negative labels are red and blue dots separately. (b), (c), (d) show prediction results made by ERM, LSSAE, and SDE-EDG respectively. The dashed line is the ground truth decision boundary of the Rotated Gaussian, which rotates 12° between two consecutive domains. Only SDE-EDG could effectively capture the exact evolving patterns, achieving the best performance on Rotated Gaussian.

# G   ADDITIONAL ABLATIONS

## G.1   IMPACT OF DIFFERENT TIME INTERVALS ON BASELINES

The experiments applying various intervals for the baselines show:

- Smaller time intervals generally enhance the performance of all methods.
- With different intervals, our method, SDE-EDG, consistently outperforms the other baselines.

Table 14: RMNIST with time interval as $\Delta t/2$

| INTERVAL= $\Delta t/2$ | 130° | 140° | 150° | 160° | 170° | 180° | AVG |
|---|---|---|---|---|---|---|---|
| ERM | 60.7 ± 0.9 | 46.6 ± 0.8 | 39.4 ± 0.8 | 38.3 ± 0.8 | 39.8 ± 0.6 | 41.0 ± 0.8 | 44.3 |
| LSSAE | 65.5 ± 0.8 | 52.6 ± 0.8 | 45.2 ± 0.8 | 39.0 ± 0.8 | 40.1 ± 0.7 | 41.1 ± 0.9 | 47.3 |
| SDE-EDG (OURS) | **75.6 ± 0.8** | **61.8 ± 0.8** | **49.9 ± 0.8** | **50.0 ± 0.9** | **45.1 ± 0.7** | **44.1 ± 0.9** | **54.4** |

Table 15: RMNIST with time interval as $2\Delta t$

| INTERVAL= $2\Delta t$ | 130° | 140° | 150° | 160° | 170° | 180° | AVG |
|---|---|---|---|---|---|---|---|
| ERM | 47.2 ± 0.8 | 37.9 ± 0.8 | 32.9 ± 0.8 | 34.9 ± 0.7 | 38.1 ± 1.0 | **43.8 ± 0.7** | 39.1 |
| LSSAE | 49.3 ± 0.9 | 38.8 ± 0.8 | 34.9 ± 0.8 | 37.8 ± 0.9 | **40.6 ± 0.8** | 40.7 ± 0.8 | 40.4 |
| SDE-EDG (OURS) | **58.6 ± 0.8** | **49.1 ± 0.7** | **45.6 ± 0.7** | **42.4 ± 0.8** | 36.9 ± 0.8 | 36.1 ± 0.8 | **44.8** |

## G.2   ABLATION OF $\alpha \in \{0.8, 1, 3, 5, 7, 9\}$

In the ablation study, the hyperparameters were varied within a wide range $\{0, 0.1, 1, 10, 100, 200\}$, causing notable fluctuations in the results. On the other hand, within the range of $\alpha \in [0.8, 10)$, the model's performance remains relatively steady (in Figure 4 (d,e)). The subsequent ablation experiments further show the stable performance by setting $\alpha = \{0.8, 1, 3, 5, 7, 9\}$:

In practice, we select hyperparameter $\alpha$ based on the validation set. Without employing cross-validation, we still can set $\alpha = 1$ yields consistent results shown in Figure 4 (d,e).

Table 16: RMNIST with $\alpha \in [0.8, 10]$

| $\alpha$ | 130° | 140° | 150° | 160° | 170° | 180° | AVG |
|---|---|---|---|---|---|---|---|
| 0.8 | $72.7 \pm 0.7$ | $59.8 \pm 0.7$ | $49.6 \pm 0.8$ | $49.2 \pm 0.7$ | $37.9 \pm 0.8$ | $39.5 \pm 0.6$ | 51.5 |
| 1 | $75.1 \pm 0.8$ | $61.3 \pm 0.9$ | $49.8 \pm 0.8$ | $49.8 \pm 0.9$ | $39.7 \pm 0.7$ | $39.7 \pm 0.9$ | 52.6 |
| 3 | $73.7 \pm 0.8$ | $58.8 \pm 0.8$ | $48.6 \pm 0.7$ | $45.9 \pm 0.7$ | $40.7 \pm 0.8$ | $39.3 \pm 0.5$ | 51.2 |
| 5 | $73.6 \pm 0.7$ | $58.1 \pm 0.8$ | $48.3 \pm 0.9$ | $42.7 \pm 0.8$ | $39.0 \pm 0.9$ | $39.7 \pm 0.7$ | 50.2 |
| 7 | $76.7 \pm 0.7$ | $63.1 \pm 0.9$ | $50.5 \pm 0.8$ | $41.8 \pm 0.9$ | $39.0 \pm 0.8$ | $39.1 \pm 0.7$ | 51.7 |
| 9 | $73.7 \pm 0.8$ | $59.8 \pm 0.7$ | $47.9 \pm 0.8$ | $44.0 \pm 0.7$ | $39.4 \pm 0.7$ | $38.5 \pm 0.6$ | 50.6 |

### G.3 ABLATION ON UNI- OR MULTI-MODAL CLASSIFICATION LOSS

We conduct the experiments with uni-modal and multi-modal classification loss respectively. The results reveal that the performance of the uni-modal approach closely resembled that of the multi-modal. The choice between the modalities often hinges upon the inherent characteristics of the datasets themselves. In real-world applications, cross-validation can be applied to determine the most suitable method for a specific dataset.

Table 17: Ablation on uni- and multi-modal loss

| DATASETS | RMNIST | PORTRAIT | POWERSUPPLY |
|---|---|---|---|
| UNI | 52.4 | 89.6 | 75.7 |
| MULTI | 52.6 | 88.9 | 75.1 |

