# OpenReview forum: "Latent Trajectory Learning for Limited Timestamps under Distribution Shift over Time"
_ICLR.cc/2024/Conference — ICLR 2024 oral_

### Official Review · Reviewer_pVBo · 2023-10-28

**Soundness:** 4 excellent
**Presentation:** 4 excellent
**Contribution:** 4 excellent
**Rating:** 8
**Confidence:** 4

**Summary:**

Considering the evolving characteristics of data distribution is crucial for machine learning models in practical applications. Such problem is formalized as Evolving Domain Generalization (EDG) in literatures. This paper a common issue in EDG where limited timestamps can lead to overfitting to source domains. In tackling this challenge, this paper introduces a novel approach that involves gathering the Infinitely Fined-Grid Evolving Trajectory (IFGET) to capture evolving dynamics and align stochastic distribution of Stochastic Differential Equations (SDEs) with IFGET. This alignment effectively captures distribution shifts across the sequence, enabling SDE-EDG to adapt the model for generalization in dynamic environments. The experimental results, conducted on various synthetic and real-world datasets, provide empirical evidence of the method's effectiveness. Overall, I believe this work proposes an innovative solution to the challenging yet under-studied problem of EDG, representing a valuable contribution to the field.

**Strengths:**

1 Infinitely Fined-Grid Evolving Trajectory (IFGET) method is novel. Capturing evolving dynamics but avoiding overfitting to limited source timestamps is achieved by collecting IFGET in the latent space using continuous interpolated samples, a strategy that has demonstrated its effectiveness in ablation experiments. Specifically, this paper proposes to collect IFGET such that the evolving patterns are learned from trajectories of the individual sample instead of collective behaviors in existing EDG methods. Therefore, IFGET provides more accurate evolving trajectories by tracking individual sample's behavior, providing a finer-grained understanding of individual sample behavior.

2 To the best of my knowledge, this is the first work to introduce Stochastic Differential Equations (SDEs) to address EDG tasks. The utilization of SDEs for modeling the continuously evolving trajectories of the latent space in the EDG problem is natural because of the inherent capabilities of SDEs in characterizing continuous stochastic processes.

3 Figures 2 and 3(3) serve as clear illustrations of the superior performance achieved by capturing evolving dynamics with IFGET, affirming its indispensable role in addressing the challenges within the EDG problem.

4 This paper is well-presented and easy to follow. The paper demonstrates the effectiveness of SDE-EDG on various datasets, including simple and complex datasets, showcasing its outperformance compared to other baseline methods by effectively capturing evolving dynamics.

**Weaknesses:**

1 I doubt the existence of evolving dynamics. It's worth considering scenarios where time-related tasks might lack evolving patterns and instead exhibit random shifts.

2 Linear interpolations may not reflect the true evolving trajectories, since the real evolving trajectories might be nonlinear and complex.

3 Considering that IFGET is an approximation, and the existence of ground truth sample-to-sample correspondence is uncertain, how do we guarantee that IFGET accurately represents the real evolving trajectories?

4 The authors conducted an ablation on weighting on IFGET; however, I am curious about the performance of IFGET without continuous interpolations, which will show whether continuous interpolations improve generalization to sparse timestamps.

5  I understand neural SDEs that neural SDEs do not rely on the assumption of the model being uni-modal, in contrast to prevelant uni-modal Gaussian distributions. However, most deep learning methods assume the classification tasks to exhibit uni-modal characteristics through using ERM classification loss. Therefore, I am wondering whether uni-modal or multi-modal classification loss contributes to the accuracy improvement of SDE-EDG in EDG.

**Questions:**

The computational costs of Neural SDEs, particularly regarding backward gradient propagation, are relatively high. However, this paper does not provide discussions of this aspect.

typos: in section 5.1, two repetitive "Ocular Disease";

In section 4.4, you should not use the same $k$ in the summation of denominator for $\frac{ |\mathbb{S}^k_m|}{\sum_{k=1}^K|\mathbb{S}^{k}_m|}$

---

> ### Author Response · Authors · 2023-11-19
> **Rebuttal from Authors of Paper2147 to Reviewer pVBo (1/2)**
>
> Thank you for taking the time to provide your insightful comments. We now address your concerns.
>
> >W1. I doubt the existence of evolving dynamics. It's worth considering scenarios where time-related tasks might lack evolving patterns and instead exhibit random shifts.
>
> A1. We appreciate the thoughtful comments.  In cases where evolving dynamics aren't present, it aligns with the traditional Domain Generalization problem that's extensively discussed and explored in existing literature [1]. On the other hand, numerous real-world applications exhibit evolving dynamics. For instance: i) An ocular disease detection system requires an acquisition of evolving patterns across different ages to accurately predict outcomes for elder patients. ii) Self-driving systems contend with latent factors that shift over time, encompassing seasonal changes, varying weather conditions, and alterations in street environments. iii) Recommendation systems employed in fashion retail, where sales are influenced by temporal factors such as trends, weather, seasons, and more. This demonstrates that latent evolving patterns are pervasive in real-world scenarios.
>
> >W2. Linear interpolations may not reflect the true evolving trajectories, since the real evolving trajectories might be nonlinear and complex.
>
> A2. Yes, there are approximating errors between the linear interpolations and real samples. To mitigate such issues,
> the interpolation rate $\lambda$ is drawn from the Beta distribution. By setting $\beta_1,\beta_2<1$ for the Beta-distributed $\lambda\sim\mathcal{B}(\beta_1,\beta_2)$, $\lambda$ tends to approach either $0$ or $1$. When $\lambda\rightarrow 1$, linear interpolation can be viewed as a first-order Taylor expansion around $\small(m+1)$ here
> $$\hat{z}=(1-\lambda) z_{m}+\lambda z_{m+1}=z_{m+1} +[(m+\lambda)-(m+1)]\frac{z_{m+1}-z_{m}}{(m+1)-m}\approx z_{m+1}+(1-\lambda) f'(m+1)$$
> where $z_m$ is parametrized by a function $f(m)$, $\frac{z_{m+1}-z_{m}}{(m+1)-m}$ approximates the first-order derivative. The aim is to minimize error by sampling $\lambda$ such that it approaches $1$, causing $(1-\lambda)$ to approach $0$. Conversely, if we expand around $m$, we tend to sample $\lambda$ closer to $0$ to minimize the approximation error. Overall, the approximating errors remain tolerable, as the interpolations show capabilities of improving learning evolving patterns and enhancing performance. Using mixup [2] as an example, an interpolation generated through a weighted sum of two samples from distinct domains doesn't exist in the real world. Despite this, it facilitates learning and functions effectively as a regularizer.
>
> >W3.  Considering that IFGET is an approximation, and the existence of ground truth sample-to-sample correspondence is uncertain, how do we guarantee that IFGET accurately represents the real evolving trajectories?
>
> A3. The existence of evolving dynamics is normally decided by prior knowledge or expert knowledge. The ground truth sample-to-sample correspondence might not exist in the collected datasets, but the efficacy of IFGET learning is validated by its contribution to the acquisition of evolving representations, as demonstrated by Figure 2 and Figure 3 (d). For instance, in datasets such as the Ocular Disease dataset, where images of the same individual at different ages might be absent, we can search the possible sample by leveraging similar features existing in behaviors or symptoms. This enables the approximation of complete latent evolving trajectories effectively, filling in the absence left in the evolving path.
>
> On the other hand, the linear interpolation, being a first-order Taylor expansion, introduces approximation errors sailing as $\mathcal{O}(\Delta t^2)$, an error that remains tolerable. Furthermore, we have computed the Mean Squared Error (MSE) between the interpolations and the real samples on the Sine dataset in the table below. Using the identity function as the backbone, this table demonstrates that the approximation error remains relatively low.
>
> Domain Interval |$(0,\frac{\pi}{24}]$    |$(\frac{\pi}{24},\frac{\pi}{12}]$|$(\frac{\pi}{12},\frac{\pi}{8}]$|$(\frac{\pi}{8},\frac{\pi}{6}]$|$(\frac{\pi}{6},\frac{5\pi}{24}]$|$(\frac{5\pi}{24},\frac{\pi}{4}]$  |$(\frac{\pi}{4},\frac{7\pi}{24}]$           |$(\frac{7\pi}{24},\frac{\pi}{3}]$           |$(\frac{\pi}{3},\frac{3\pi}{8}]$           |$(\frac{3\pi}{8},\frac{5\pi}{12}]$           |$(\frac{5\pi}{12},\frac{\pi}{2}]$           |
> |:-------:|:-------:|:----------------------------:|:--------------------------:|:--------------------------:|:--------------------------:|:--------------------------:|:--------------------------:|:--------------------------:|:----------------------------:|:--------------------------:|:--------------------------:|
> |MSE|0.352|0.401|0.125|0.210|0.302|0.148|0.136|0.167|0.195|0.371|0.531|

---

> ### Author Response · Authors · 2023-11-19
> **Rebuttal from Authors of Paper2147 to Reviewer pVBo (2/2)**
>
> >W4. The authors conducted an ablation on weighting on IFGET; however, I am curious about the performance of IFGET without continuous interpolations, which will show whether continuous interpolations improve generalization to sparse timestamps.
>
> A4. We experiment with or without the interpolations utilized per iteration to assess their impact on RMNIST:
>
>
> |           |130$^\circ$    |140$^\circ$|150$^\circ$|160$^\circ$|170$^\circ$|180$^\circ$  | AVG             |
> |:-------:|:----------------------------:|:--------------------------:|:--------------------------:|:--------------------------:|:--------------------------:|:--------------------------:|:--------------------------:|
> |w/o interpolation|69.6 $\pm$ 0.7 | 53.3 $\pm$ 0.9 | 47.7 $\pm$ 0.8 | 45.4 $\pm$ 0.8 | 39.8 $\pm$ 0.8 | 41.3 $\pm$ 0.7| 49.5 |
> |w/ interpolation|75.1 $\pm$ 0.8|61.3 $\pm$ 0.9|49.8 $\pm$ 0.8|49.8 $\pm$ 0.9|39.7 $\pm$ 0.7|39.7 $\pm$ 0.9| 52.6|
>
> The interpolations significantly improve the performance.
>
> > W5. I understand neural SDEs that neural SDEs do not rely on the assumption of the model being uni-modal, in contrast to prevelant uni-modal Gaussian distributions. However, most deep learning methods assume the classification tasks to exhibit uni-modal characteristics through using ERM classifi cation loss. Therefore, I am wondering whether uni-modal or multi-modal classification loss contributes to the accuracy improvement of SDE-EDG in EDG.
>
> A5. We conduct the experiments with uni-modal and multi-modal classification loss respectively. The results reveal that the performance of the uni-modal approach closely resembled that of the multi-modal. The choice between the modalities often hinges upon the inherent characteristics of the datasets themselves. In real-world applications, employing cross-validation can be applied to determine the most suitable method to a specific dataset.
>
> |       Datasets |RMNIST    |Portrait       | PowerSupply
> |:-------:|:----------------------------:|:--------------------------:|:--------------------------:|
> |Uni|52.4 | 89.6 | 75.7|
> |Multi|52.6|88.9|75.1
>
> We have added this experimental result to our appendix.
>
> > Q. The computational costs of Neural SDEs, particularly regarding backward gradient propagation, are relatively high. However, this paper does not provide discussions of this aspect.
>
> A. We appreciate the thoughtful comments. The computational efficiency of the Vanilla SDE method can be time-consuming, but our implementation utilizing the "torchsde" package, as in [1], significantly saves computational resources. According to [1], the computational complexity of the neural SDE component in the SDE-EDG method is characterized by $\mathcal{O}\Big((M+L) D\Big)$, where $M$ is the number of source domains, $L$ is the number of target domains, and $D$ accounts for the number of parameters within the drift and diffusion neural networks.
>
> Regarding the backward gradient propagation, "torchsde" leverages the adjoint method for gradient computation, for which an exact time complexity analysis is unavailable in literature to the best of our knowledge. We conduct experiments and record the runtime per iteration during the training phase. The below results indicate that SDE-EDG is not the fastest computational time among the considered methods. However, it's noted that SDE-EDG consistently outperforms both baselines, achieving an average accuracy improvement of at least 11.6% across all datasets.
>
>
>
>
> |       Algos |DDA    |GI       | SDE-EDG
> |:-------:|:----------------------------:|:--------------------------:|:--------------------------:|
> |RMNIST|0.19s | 2.80s | 0.27s|
> |Portrait|0.77s|7.48s|0.31s
>
> >Typos
>
> Thanks. We have fixed these typos in the revisions.
>
> [1] Wang, Jindong, et al. "Generalizing to unseen domains: A survey on domain generalization." _IEEE Transactions on Knowledge and Data Engineering_ (2022).
> [2] Xu, Minghao, et al. "Adversarial domain adaptation with domain mixup." _Proceedings of the AAAI conference on artificial intelligence_. Vol. 34. No. 04. 2020.

---

### Official Review · Reviewer_y1vq · 2023-10-29

**Soundness:** 3 good
**Presentation:** 3 good
**Contribution:** 3 good
**Rating:** 8
**Confidence:** 3

**Summary:**

This paper proposes a SDE-EDG method in order to solve the challenges that limited number of timestamps are available in evolving domain generalization problem via collecting infinite fine-grid evolving trajectory of the data distribution with continuous-interpolated samples to bridge temporal gaps.

**Strengths:**

- The writting of the paper is good.
- The problem studied in this paper is well and clearly formulated.
- The illustration and demonstration is convincing and solid.
- The proposed SDE-EDG achieves better empirical results on several benchmark datasets than existing SOTA methods.

**Weaknesses:**

- The illustration of continuous-interpolated samples is not very clear. I am a little bit confused about why the samples generated with linear interpolation method are called "continuous".

**Questions:**

1. What are the differences between continuous-interpolated samples and samples used in previous works?

2. From my perspective, the interpolated samples are generated discretely. Thus, does the SDE-EDG method approximates the continuous case in the way of discretely sampling?

3. According to my understanding, Eq.(7) aims at learn a set of parameters for $f_k$ and $g_k$ so that the learned models can mimic the approximated trajectory of data. In my opinion, such operation relies heavily on the quality of data representations, especicially at the early phase of training, since the feature extractor $\phi$ is not well trained yet. Thus, in such case, will the linear interpolation results in some "bad interpolation"?

4. According to the ablation study, the hyper-parameter $\alpha$ seems sensitive for different datasets. Since $\alpha$ scale the magnitude of $\mathcal{L}_{mle}$ that plays the role of regularizer, could you explain such phenomenon?

---

> ### Author Response · Authors · 2023-11-19
> **Rebuttal from Authors of Paper2147 to Reviewer y1vq (1/2)**
>
> We thank your time and insightful comments. We now address your concerns.
>
> > W. The illustration of continuous-interpolated samples is not very clear. I am a little bit confused about why the samples generated with the linear interpolation method are called "continuous".
>
> A. We call it "continuous" because: For each interpolation, $\hat{z}=(1-\lambda)z_m+\lambda z_{m+1}$, $\lambda$ is sampled from a Beta distribution. Thus, $\lambda$ is expected to be any value between $(0,1)$ with possibility and hence it allow us to approximate any time moment between the $m$-th and $(m+1)$-th timestamps. We have made it clearer in the revisions (above Section 4.2).
>
> > Q1. What are the differences between continuous-interpolated samples and samples used in previous works?
>
> A1. Previous works use samples from a limited number ($M$) of source domains to train the temporal model (using LSTM [1] or temporal transition model [2]), potentially leading to overfitting issues. To tackle this issue, we propose the use of interpolated samples, creating a finer-grid latent trajectory, which facilitates the learning of evolving patterns.
>
> >Q2. From my perspective, the interpolated samples are generated discretely. Thus, does the SDE-EDG method approximates the continuous case in the way of discretely sampling?
>
> A2. Yes. As mentioned in (A) above, given that the interpolation coefficient $\lambda$ ranges between $0$ and $1$, we consider the possibility of sampling data from any timestamp within the interval $[0,T]$. This enables us to view the trajectory as continuous.
>
> >Q3. According to my understanding, Eq.(7) aims at learning a set of parameters for $f_k$ and $g_k$ so that the learned model can mimic the approximated trajectory of data. In my opinion, such an operation relies heavily on the quality of data representations, especially at the early phase of training, since the feature extractor is not well trained yet. Thus, in such a case, will the linear interpolation result in some "bad interpolation"?
>
> A3. We appreciate your valuable comments. During the initial training stage, the interpolation unavoidably might not be precise approximations. However, as the training goes on, the model is updated and the representations gradually converge to optimal representations. This training process involves mutual interactions: The IFGET acts as a regularizer, preserving evolving information within the representations, as illustrated in Figure 2 and Figures 3 (d-e); Simultaneously, the representations contained the evolving dynamics in turn facilitate the learning of $f_k$ and $g_k$.  In typical classification tasks, initial representations may not be optimal but are still utilized for training the classification head (often an MLP). Eventually, all components in SDE-EDG converge towards an optimum due to the benefits of the end-to-end training protocol.

---

> ### Author Response · Authors · 2023-11-19
> **Rebuttal from Authors of Paper2147 to Reviewer y1vq (2/2)**
>
> >Q4. According to the ablation study, the hyper-parameter seems sensitive for different datasets. Since scale the magnitude of that plays the role of regularizer, could you explain such phenomenon?
>
> A4. In the ablation study, the hyperparameters were varied within a wide range $\{0, 0.1, 1, 10, 100, 200\}$, causing notable fluctuations in the results. On the other hand, within the range of $\alpha \in [1, 10]$, the model's performance remains relatively steady (in Figure 4 (d,e)). The subsequent ablation experiments on RMNIST further shows the stable performance by setting $\alpha = \{0.8, 1, 3, 5, 7, 9\}$:
> |       $\alpha$        |130$^\circ$    |140$^\circ$|150$^\circ$|160$^\circ$|170$^\circ$|180$^\circ$  | AVG             |
> |:-------:|:----------------------------:|:--------------------------:|:--------------------------:|:--------------------------:|:--------------------------:|:--------------------------:|:--------------------------:|
> |0.8|72.7 $\pm$ 0.7 | 59.8 $\pm$ 0.7 | 49.6 $\pm$ 0.8 | 49.2 $\pm$ 0.7 | 37.9 $\pm$ 0.8 | 39.5 $\pm$ 0.6| 51.5 |
> |1|75.1 $\pm$ 0.8|61.3 $\pm$ 0.9|49.8 $\pm$ 0.8|49.8 $\pm$ 0.9|39.7 $\pm$ 0.7|39.7 $\pm$ 0.9| 52.6|
> |3|73.7 $\pm$ 0.8 | 58.8 $\pm$ 0.8 | 48.6 $\pm$ 0.7 |45.9 $\pm$ 0.7 | 40.7 $\pm$ 0.8 | 39.3 $\pm$ 0.5| 51.2|
> |5|73.6 $\pm$ 0.7 | 58.1 $\pm$ 0.8 | 48.3 $\pm$ 0.9 | 42.7 $\pm$ 0.8| 39.0 $\pm$ 0.9 | 39.7 $\pm$ 0.7 | 50.2
> |7|76.7 $\pm$ 0.7| 63.1 $\pm$ 0.9 |50.5 $\pm$ 0.8 | 41.8 $\pm$ 0.9 | 39.0 $\pm$ 0.8 | 39.1 $\pm$ 0.7|51.7|
> |9       |73.7 $\pm$ 0.8|59.8 $\pm$ 0.7|47.9 $\pm$ 0.8|44.0 $\pm$ 0.7| 39.4 $\pm$ 0.7| 38.5 $\pm$ 0.6| 50.6|
>
> In practice, we select hyperparameter $\alpha$ based on the validation set. Without employing cross-validation, we still can set $\alpha = 1$ yields consistent results shown in Figure 4 (d,e). We have added this experimental result to our appendix.
>
> [1] Bai, Guangji, Chen Ling, and Liang Zhao. "Temporal Domain Generalization with Drift-Aware Dynamic Neural Networks." _The Eleventh International Conference on Learning Representations_. 2022.
> [2] Qin, Tiexin, Shiqi Wang, and Haoliang Li. "Generalizing to Evolving Domains with Latent Structure-Aware Sequential Autoencoder." _International Conference on Machine Learning_. PMLR, 2022.

---

> ### Comment · Reviewer_y1vq · 2023-11-22
> **Response to rebuttal from authors**
>
> Dear authors,
>
> Thank you for your responses. Basically, I think my concerns are addressed.
>
> Thus, I would like to maintain my score and vote for accepting this paper.
>
> Best wishes,
>
> Reviewer y1vq

---

> > ### Author Response · Authors · 2023-11-22
> >
> > We are glad to see that the reviewer's concerns have been addressed. Once more, we appreciate the time and effort the reviewer dedicated to reviewing our paper and offering valuable feedback.

---

### Official Review · Reviewer_KuA7 · 2023-10-31

**Soundness:** 3 good
**Presentation:** 3 good
**Contribution:** 3 good
**Rating:** 8
**Confidence:** 1

**Summary:**

This paper investigates the evolving domain generalization (EDG) task. Previous works have difficulties in capturing evolving dynamics due to limited timestamps. To address this, this paper simulates the data distribution with continuous-interpolated samples and leverages SDEs to capture evolving distribution trends. The proposed method achieves SOTA performance on all 9 benchmarks.

**Strengths:**

1. The proposed method achieves state-of-the-art performance on the benchmarks.

2. Extensive experiments and analysis demonstrate the effectiveness of the proposed method.

**Weaknesses:**

1. In one iteration, for two consecutive domains, only one interpolated sample is generated. Can more than one sample be generated and used? For example, we can first generate one interpolated sample via Eq.5 and then set this sample as $\tilde{z}_{m+1}$ to generate the second sample. If so, how does this influence the performance?

2. Table 2 shows that smaller temporal gaps improve the generalization ability. How is the improvement of SDE-EDG over the baseline when using different time intervals?

**Questions:**

See weaknesses.

---

> ### Author Response · Authors · 2023-11-19
> **Rebuttal from Authors of Paper2147 to Reviewer KuA7**
>
> Thank you for your time and valuable comments. We now address your concerns.
>
> >Q1. In one iteration, for two consecutive domains, only one interpolated sample is generated. Can more than one sample be generated and used? For example, we can first generate one interpolated sample via Eq.5 and then set this sample as $\tilde{z}_{m+1}$ to generate the second sample. If so, how does this influence the performance?
>
>
>
> A1. Sorry for causing the misunderstanding.  We have clarified this in the Equation 5 of the revisions. In practice, we generate an interpolation for each sample pair within the sampled batch of size $N_B$, resulting in $N_B$ interpolations per iteration.
>
> Interpolating twice to generate the second sample is equivalent to generating the interpolation once. Specifically, if we denote the second sample resulting from interpolation twice as:
>
> $${\hat z_{twice}}=( 1 - \lambda_2) z_m+ \lambda_2 {\hat z_{m+1}} =(1-\lambda_2)z_{m}+ \lambda_2 [(1-\lambda_1) z_m+\lambda_1 z_{m+1}] =(1-\lambda_1\lambda_2)z_m+\lambda_1\lambda_2 z_{m+1}$$
>
> where $\lambda_1$ and $\lambda_2$ are two interpolation rates for the first and second interpolated samples. Taking $\lambda_1\lambda_2=\lambda_3$, interpolating twice to generate the second sample is equivalent to generating the first sample with an interpolation rate of $\lambda_3$.
>
>
>
>
> >Q2. Table 2 shows that smaller temporal gaps improve the generalization ability. How is the improvement of SDE-EDG over the baseline when using different time intervals?
>
> The experiments applying various intervals on RMNIST for the baselines show: 1. Smaller time intervals generally enhance the performance of all methods. 2. With different intervals, our method, SDE-EDG, consistently outperforms the other baselines.
> |  interval= $\Delta t/2$           |130$^\circ$    |140$^\circ$|150$^\circ$|160$^\circ$|170$^\circ$|180$^\circ$  | AVG             |
> |:-------:|:----------------------------:|:--------------------------:|:--------------------------:|:--------------------------:|:--------------------------:|:--------------------------:|:--------------------------:|
> |ERM|60.7 $\pm$ 0.9 | 46.6 $\pm$ 0.8 | 39.4 $\pm$ 0.8| 38.3 $\pm$ 0.8 | 39.8 $\pm$ 0.6 | 41.0 $\pm$ 0.8| 44.3 |
> |LSSAE|65.5 $\pm$ 0.8|52.6 $\pm$ 0.8|45.2 $\pm$ 0.8|39.0 $\pm$ 0.8|40.1 $\pm$ 0.7|41.1 $\pm$ 0.9| 47.3|
> |SDE-EDG (ours)|__75.6 $\pm$ 0.8__|**61.8 $\pm$ 0.8**|**49.9 $\pm$ 0.8**|**50.0 $\pm$ 0.9**|**45.1 $\pm$ 0.7**|**44.1 $\pm$ 0.9** | __54.4__|
>
>
> |  interval= $2\Delta t$           |130$^\circ$    |140$^\circ$|150$^\circ$|160$^\circ$|170$^\circ$|180$^\circ$  | AVG             |
> |:-------:|:----------------------------:|:--------------------------:|:--------------------------:|:--------------------------:|:--------------------------:|:--------------------------:|:--------------------------:|
> |ERM|47.2 $\pm$ 0.8 | 37.9 $\pm$ 0.8 | 32.9 $\pm$ 0.8| 34.9 $\pm$ 0.7 |38.1 $\pm$ 1.0 | __43.8 $\pm$ 0.7__| 39.1 |
> |LSSAE|49.3 $\pm$ 0.9|38.8 $\pm$ 0.8|34.9 $\pm$ 0.8|37.8 $\pm$ 0.9 |__40.6 $\pm$ 0.8__|40.7 $\pm$ 0.8| 40.4|
> |SDE-EDG (ours) |__58.6 $\pm$ 0.8__|__49.1 $\pm$ 0.7__|__45.6 $\pm$ 0.7__|__42.4 $\pm$ 0.8__|36.9 $\pm$ 0.8|36.1 $\pm$ 0.8|__44.8__|
>
> We have added the experimental results to our appendix.

---

> > ### Comment · Reviewer_KuA7 · 2023-11-22
> > **Response to the rebuttal**
> >
> > Dear authors,
> >
> > Thanks for the response. My concerns are addressed and I will increase my rating to accept.

---

> > > ### Author Response · Authors · 2023-11-22
> > >
> > > We are delighted to see that the concerns raised by the reviewer have been successfully addressed. We want to express our sincere gratitude to the reviewer for dedicating their time and effort to thoroughly examine our paper and offer invaluable feedback.

---

### Official Review · Reviewer_bSwA · 2023-11-06

**Soundness:** 3 good
**Presentation:** 2 fair
**Contribution:** 3 good
**Rating:** 8
**Confidence:** 3

**Summary:**

The present work proposes SDE-EDG, a novel learning approach for solving the problem of Evolving Domain Generalization in the context of (neural) Stochastic Differential Equations. Real-world applications pose challenges to existing EDG approaches, such as inconsistent data availability over time, which leads to difficulties in capturing the underlying dynamics, risks overfitting and reduces the quality of generalization to unseen data. Further, distributions that change over time cannot be taken into account as they contradict the stationarity assumption. The authors claim to overcome all those limitations by incorporating the learning of latent trajectories that describe the distributional changes over time. The latter is done using variational inference in the setting of neural SDE approaches and by improving latent trajectory fits by bridging temporal gaps through linear interpolation of samples from consecutive distributions (the IFGET module). Empirical evidence is provided through synthetical and real world experiments demonstrating superior performance compared to existing state-of-the-art methods.

**Strengths:**

The generative approach is not only convincing due to its quantitative superiority, but also provides insights into the black-box mechanisms of model learning, which can be easily generalized to unknown target areas. The authors exemplify how such investigation of representation in the form of latent trajectories (e.g., see Figure 2 and 3(d/e)) provides valuable insights. Further, empirical evidence is accessed through experiments on synthetical and real-world data as well as reproducibility is granted by submitted code files. The paper content is original to the best of my knowledge and, with few exceptions, has no spelling or grammatical flaws. Finally, on a positive note, the authors perform a series of ablation studies to test the validity of their claim that the proposed IFGET module contributes complementary information to learning evolving dynamics in a classical neural SDE scheme.

**Weaknesses:**

1. Learning neural SDEs is traditionally successfully presented in the framework of variational Bayesian inference (e.g. Li et al. 2020), where stochastic gradient descent methods are applied to minimize the evidence lower bound (ELBO). I wonder why the authors do not clearly address the reference to variational Inference in the manuscript, especially since the indications (e.g., the graphical model in Figure 1, derivation of the likelihood loss of IFGET in Lemma B.1) are given.

2. The Sine experiment (Figure 3) is arguably a rather simple problem from a dynamic point of view and yet already indicates the limited extrapolation quality of the approach. Of course, this reduces my confidence in the generalization ability of the approach. I suspect possible causes in (i) an unfavorable modeling of drift and diffusion coefficients and (ii) that the linear interpolation approach in the IFGET module is too coarse. Can you comment on this?

3. The IGET module reminds me on a approach by Kidger et al. 2020 (Neural Controlled Differential Equations for Irregular Time Series) in which the authors also try to improve the learning of latent trajectories by including interpolated sample trajectories, in their terms a "controlled path”. This is done in the ODE setting, but can be integrated into the SDE setting if the approach is applied to learning the drift coefficient. Can you explain in more detail how your approach differs?

**Questions:**

1. (p.3) Do you assume all source domains have the same sample size $N$ or can it vary?
2. (p.4) You repeat here (unnecessarily, in my opinion) the argument of "sample complexity", which already appeared on p. 1. However, I would like to ask if you could explain this argument in more detail?
3. (p.4) Can you define $N_B$?
4. (p.4) Why do you choose to draw the interpolation rate $\lambda$ from a Beta distribution?
5. How many interpolation points are generated between two consecutive data points, i.e., how many $\lambda 's$ are drawn; am I right in assuming that only one is drawn? Would an increase in the number of intermediate points result in better performance?
6. Neural SDEs are known to be very resource consuming. Can you assess the runtime complexity, for example?
7. (Figure 4(c)) Can you assess what happens at Domain Index 29; why does SDE-EDG show a reduction in performance?

Minor:
    - Redundancy in the references.

**Details Of Ethics Concerns:**

--

---

> ### Author Response · Authors · 2023-11-19
> **Rebuttal from Authors of Paper2147 to Reviewer bSwA (1/3)**
>
> Thank you for your time and valuable feedback. We now address your concerns.
>
> > W1. Learning neural SDEs is traditionally successfully presented in the framework of variational Bayesian inference (e.g. Li et al. 2020), where stochastic gradient descent methods are applied to minimize the evidence lower bound (ELBO). I wonder why the authors do not clearly address the reference to variational Inference in the manuscript, especially since the indications (e.g., the graphical model in Figure 1, derivation of the likelihood loss of IFGET in Lemma B.1) are given.
>
> A1. Thank you for your insightful comments. We admit in SDE models, variational inference is commonly employed, allowing the assumption of a prior distribution for the diffusion term, such as the Ornstein–Uhlenbec (OU) process or Wiener process. This serves as a regularization technique, mitigating overfitting by regularizing the diffusion term. We also highlighted this in the final line of page 2 within Section 2 (Related Works) and we revised the last sentence of Section 2 to stress this point.
>
> On the other hand, we also note that variational inference is not a mandatory requirement for optimizing SDE models [2, 3, 4]. Maximum likelihood, a widely used method, is also applicable for optimizing SDEs [2, 4]. In SDE-EDG method, we indeed adopted maximum likelihood rather than variational inference.
>
> The graphical models depicted in Figure 1 of our paper and Figure 4 in [1] serve similar purposes. Their roles are not related to variational inference but rather illustrate the generative process of the data. Lemma B.1 also shows the derivation of maximum likelihood, not the ELBO in variational inference.
>
> >W2. The Sine experiment (Figure 3) is arguably a rather simple problem from a dynamic point of view and yet already indicates the limited extrapolation quality of the approach. Of course, this reduces my confidence in the generalization ability of the approach. I suspect possible causes in (i) unfavorable modeling of drift and diffusion coefficients and (ii) that the linear interpolation approach in the IFGET module is too coarse. Can you comment on this?
>
> A2. We appreciate the valuable comment. The cause is not modeling or interpolation method, but the EDG setup of the Sine dataset in our experiments making the task very challenging. We follow the setup of LSSAE [5], where we divide a complete sine $[0,2\pi]$ into 24 domains, each spanning $\frac{\pi}{12}$. Most importantly, sine in EDG experiments only has 12 domains $[0,\pi)$ for training, $[\pi, \frac{4\pi}{3})$ for validation, $[\frac{4\pi}{3},2\pi]$ for reporting test results. This means we train the model using only half of the sine instead of the entire period of the sine curve, which makes the task significantly more challenging.
>
> From Table 6 in the appendix, the results of Sine show that under this setting, none of the methods can generalize well to unseen domains, especially when the test domains are far away from the training domains. However, our method SDE-EDG shows performance improvement by 0.8% compared to the best baseline.
>
> In Figure 3(d), where we illustrate the latent paths learned by the SDE component, we set the backbone as the identity function and the hidden dimension to $2$ for better visualization compared to the original Sine. However, for the experiments reported in Table 1, we apply a hidden dimension of $32$, as indicated in Table 12 of the appendix. This leads to better fitting capabilities and better generalization than what Figure 3(d) displays especially for the test domains that are close to the training domains.

---

> ### Author Response · Authors · 2023-11-19
> **Rebuttal from Authors of Paper2147 to Reviewer bSwA (2/3)**
>
> >W3. The IFGET module reminds me on a approach by Kidger et al. 2020 (Neural Controlled Differential Equations for Irregular Time Series) in which the authors also try to improve the learning of latent trajectories by including interpolated sample trajectories, in their terms a "controlled path”. This is done in the ODE setting, but can be integrated into the SDE setting if the approach is applied to learning the drift coefficient. Can you explain in more detail how your approach differs?
>
> A3.  We highlight distinctions between [6] and our approach SDE-EDG in four aspects.
>
> 1. Interpolation is applied for different purposes: [6] aims to obtain gradients via interpolations, but we aim to obtain augmentations from unseen timestamps. More specifically, [6] employing a spline method to approximate the trajectories of $X$, and computing gradients from the derivative of the spline function. In contrast, SDE-EDG utilizes interpolations as augmentations to enhance the learning of latent trajectories, aiming to mitigate overfitting to limited timestamps.
>
> 2. Different problem scope: [6] focuses on predicting the labels $Y$ based on the whole time series $( X_1,\ldots,X_M)$, which is a time-series classification task. Conversely, in EDG setup, SDE-EDG aims to utilize latent temporal patterns learned from $\(X_t,Y_t)$, $t=1,\ldots,M$ collected from the seen timestamps to improve predictions of $Y$ based on observations of $X$ in future timestamps.
>
> 3. Interpolate in different spaces: [6] utilizes interpolation on $X$ within the original space, whereas SDE-EDG employs interpolation on $Z$ within the latent space. The choice of interpolating $Z$ offers potential advantages in capturing the underlying evolving dynamics of the data generation process compared to the original observations $X$.
>
> 4. We adopt different interpolation techniques: SDE-EDG uses direct linear interpolation to reduce time computational complexity, while [6] employs cubic spline interpolations.
>
> Additionally, our novelty doesn't lie in the interpolation method itself; rather, it lies in addressing overfitting to sparse timestamps through interpolations and effectively capturing evolving dynamics by constructing IFGET under EDG settings.
>
> >Q1. (p.3) Do you assume all source domains have the same sample size $N$ or can it vary?
>
> A1. For simplicity in notation, we use the same symbol $N$ for source domains. Empirically, this value can actually vary in each domain. In other words, we do not assume the same sample size $N$ in our algorithm implementation.
>
> >Q2. (p.4) You repeat here (unnecessarily, in my opinion) the argument of "sample complexity", which already appeared on p. 1. However, I would like to ask if you could explain this argument in more detail?
>
> A2. Thanks for your valuable feedback. We have deleted this sentence in Section 4.1 in revisions. We use the sample complexity as a support to our motivation. The sample complexity, with a smaller number of timestamps $M$, leads to **a looser generalization bound**, showed by the complexity term $\mathcal{O}(\sqrt{1/M})$. This indicates that **overfitting occurs when the model learns from a limited number of timestamps**. In light of this, our proposed IFGET involves interpolation between timestamps to generate approximations that act as augmentation from different unseen intermediate timestamps.
>
> >Q3. (p.4) Can you define $N_B$?
>
> A3. $N_B$ denotes the number of samples for a batch in a training iteration. We define it below the Equation 4.
>
> > Q4. (p.4) Why do you choose to draw the interpolation rate $\lambda$ from a Beta distribution?
>
> A4. Beta-distribution is verified for linear interpolation in literature [8, 9]. In SDE-EDG, $\lambda$ thereby will have higher chances to be close to either $0$ or $1$ by sampling from Beta distribution, leading to a more accurate approximation. By choosing $\beta_1,\beta_2<1$ of Beta-distribution $\lambda\sim\mathcal{B}(\beta_1,\beta_2)$, $\lambda$ is closer to $0$ or $1$. When $\lambda\rightarrow 1$, we can regard linear interpolation as the first order Taylor expansion at $\small(m+1)$ here
> $$\hat{z}=(1-\lambda) z_{m}+\lambda z_{m+1}=z_{m+1} +[(m+\lambda)-(m+1)]\frac{z_{m+1}-z_{m}}{(m+1)-m}\approx z_{m+1}+(1-\lambda) f'(m+1)$$
> where $z_m$ is parametrized by a function $f(m)$,  $\frac{z_{m+1}-z_{m}}{(m+1)-m}$ approximates the first-order derivative. Since it is an approximation, we aim to reduce the approximation error by sampling $\lambda\rightarrow 1$ and thus $(1-\lambda)\rightarrow 0$. On the other hand, if we expand on $m$, we tend to sample $\lambda$ closer to $0$ to minimize the approximation error.

---

> ### Author Response · Authors · 2023-11-19
> **Rebuttal from Authors of Paper2147 to Reviewer bSwA (3/3)**
>
> >Q5. How many interpolation points are generated between two consecutive data points, i.e., how many $\lambda$ are drawn; am I right in assuming that only one is drawn? Would an increase in the number of intermediate points result in better performance?
>
> Yes, we generate one interpolation in each training iteration and utilize a single $\lambda$ to maintain computational efficiency through batch operations on CUDA. While employing interpolations helps mitigate overfitting, it's essential to find a balanced tradeoff. Increasing the number of interpolations may introduce accumulated approximation errors between interpolated and real samples, causing a decline in performance. This claim is substantiated by the experimental results on RMNIST presented below. We vary the number of interpolations (0&#8594;1&#8594;3) applied in each iteration to assess its impact.
>
>
> |        Num of Interp        |130$^\circ$    |140$^\circ$|150$^\circ$|160$^\circ$|170$^\circ$|180$^\circ$  | AVG             |
> |:-------:|:----------------------------:|:--------------------------:|:--------------------------:|:--------------------------:|:--------------------------:|:--------------------------:|:--------------------------:|
> |0|69.6 $\pm$ 0.7 | 53.3 $\pm$ 0.9 | 47.7 $\pm$ 0.8 | 45.4 $\pm$ 0.8 | __39.8 $\pm$ 0.8__ | __41.3 $\pm$ 0.7__| 49.5 |
> |1|__75.1 $\pm$ 0.8__|61.3 $\pm$ 0.9|49.8 $\pm$ 0.8|__49.8 $\pm$ 0.9__|39.7 $\pm$ 0.7|39.7 $\pm$ 0.9| __52.6__|
> |2|74.2 $\pm$ 0.7 | __61.4 $\pm$ 0.8__ | __50.8 $\pm$ 0.9__|47.0 $\pm$ 0.8 | 37.7 $\pm$ 0.8 | 38.7 $\pm$ 0.7| 51.6|
> |3|74.5 $\pm$ 0.7 | 61.3 $\pm$ 0.9 | 48.2 $\pm$ 0.8 | 45.9 $\pm$ 0.8| 36.7 $\pm$ 0.8 | 37.8 $\pm$ 0.7 | 50.7
>
> This suggests the importance of maintaining a balanced ratio between real samples and interpolations, ideally at 1:1 to 1:2. Otherwise, the model might prioritize fitting the approximations over the real samples. We have added the experimental results to our appendix.
>
> >Q6. Neural SDEs are known to be very resource consuming. Can you assess the runtime complexity, for example?
>
> Yes, Vanilla SDE can be time-consuming. In our implementation, to improve efficiency and reduce the computational time, we adopt "torchsde" package from [1]. According to [1], the computational complexity of the neural SDE component in SDE-EDG method is $\mathcal{O}\Big((M+L) D\Big)$, where $M$ is the number of source domains, $L$ is the number of target domains, and $D$ is the number of parameters of the drift and diffusion neural networks.
>
> We conduct experiments and record the runtime per iteration during the training phase. The below results indicate that SDE-EDG is comparable among the considered methods, and both our SDE-EDG and DDA are much faster than GI. On the other hand, we can not ignore that SDE-EDG consistently outperforms both baselines, achieving an average accuracy improvement of at least 11.6% across all datasets.
> |       Algos |DDA    |GI       | SDE-EDG
> |:-------:|:----------------------------:|:--------------------------:|:--------------------------:|
> |RMNIST|__0.19s__ | 2.80s | 0.27s|
> |Portrait|0.77s|7.48s|__0.31s__
>
> We have added the experimental results to our appendix.
>
> >Q7. (Figure 4c) Can you assess what happens at Domain Index 29; why does SDE-EDG show a reduction in performance?
>
> A7. The performance tends to decrease for domains that are further away from the source domains due to the accumulation of approximation errors. Since the domain is distant from the source domains, the learned evolving patterns are less reliable.
>
>
>
> [1] Li, Xuechen, et al. "Scalable gradients for stochastic differential equations." _International Conference on Artificial Intelligence and Statistics_. PMLR, 2020.
>
> [2] Jia, Junteng, and Austin R. Benson. "Neural jump stochastic differential equations." _Advances in Neural Information Processing Systems_ 32 (2019).
>
> [3] Kidger, Patrick, et al. "Neural sdes as infinite-dimensional gans." _International conference on machine learning_. PMLR, 2021.
>
> [4] Song, Yang, et al. "Maximum likelihood training of score-based diffusion models." _Advances in Neural Information Processing Systems_ 34 (2021): 1415-1428.
>
> [5] Qin, Tiexin, Shiqi Wang, and Haoliang Li. "Generalizing to Evolving Domains with Latent Structure-Aware Sequential Autoencoder." _International Conference on Machine Learning_. PMLR, 2022.
>
> [6] Kidger, Patrick, et al. "Neural controlled differential equations for irregular time series." _Advances in Neural Information Processing Systems_ 33 (2020): 6696-6707.
>
> [7] Xu, Winnie, et al. "Infinitely deep bayesian neural networks with stochastic differential equations." _International Conference on Artificial Intelligence and Statistics_. PMLR, 2022.
>
> [8] Xu, Minghao, et al. "Adversarial domain adaptation with domain mixup." _Proceedings of the AAAI conference on artificial intelligence_. Vol. 34. No. 04. 2020.
>
> [9] Yan, Shen, et al. "Improve unsupervised domain adaptation with mixup training." _arXiv preprint arXiv:2001.00677_ (2020).

---

> > ### Comment · Reviewer_bSwA · 2023-11-21
> >
> > I thank the authors very much for their thorough response and for the detailed discussion
> > of my questions and concerns.
> >
> > You have clarified things, and I will increase my score correspondingly, under the assumption that these clarifications will make it into the updated version of the paper.
> >
> > Thank you again.

---

> > > ### Author Response · Authors · 2023-11-21
> > >
> > > We thank you for raising the score and also for your valuable and constructive feedback, which greatly improves the quality of our paper. We will add all these clarifications into revisions based on your insightful comments.

---

### Comment · Area_Chair_8pBW · 2023-11-21
**[Time Sensitive, ICLR24] Please read the authors' responses and try to discuss the remaining concerns with the authors**

Dear Reviewers,

The authors have provided detailed responses to your comments.

Could you have a look and try to discuss the remaining concerns with the authors? The reviewer-author discussion will end in two days.

We do hope the reviewer-author discussion can be effective in clarifying unnecessary misunderstandings between reviewers and the authors.

Best regards,

Your AC

---

### Meta-Review · Area_Chair_8pBW · 2023-12-05

**Metareview:**

Evolving domain generalization (EDG) is an important problem because the distribution changes over time in many scenarios. This paper tracks this important problem and proposes a new perspective (from SDEs) to solve this problem. All reviewers agree that this new perspective is interesting to the field and has the potential to influence more work in the future, based on solid experiments and strong motivations. More importantly, theoretical contributions are included in this paper as well.

**Justification For Why Not Higher Score:**

This paper is suggested as Accept with Oral because this paper provides a new perspective to an important problem setting and insightful theoretical results, which would motivate more papers in this field in the future.

**Justification For Why Not Lower Score:**

This paper is suggested as Accept with Oral because this paper provides a new perspective to an important problem setting, which would motivate more papers in this field in the future. All reviewers agree with this contribution that should be considered as an Oral paper.

---

### Decision · Program_Chairs · 2024-01-16

Accept (oral)